# How flat is flat? Investigating snow topography and the spatial variability of snow surface temperature on landfast sea ice using UAVs in McMurdo Sound, Antarctica

Julia Martin<sup>1,2</sup>, Ruzica Dadic<sup>1,2</sup>, Brian Anderson<sup>1</sup>, Roberta Pirazzini<sup>3</sup>, Oliver Wigmore<sup>1</sup>, and Lauren Vargo<sup>1</sup>

**Correspondence:** Julia Martin (julia.martin@vuw.ac.nz)

**Abstract.** How do snow distribution patterns influence the surface temperature of snow on sea ice? Despite its crucial role in the sea-ice energy balance, snow on Antarctic sea ice remains under-sampled and poorly understood. In our study, we combined Uncrewed Aerial Vehicle (UAV) and ground-based measurements to obtain high resolution  $(9\,\text{cm/px})$  maps of snow topography (Digital Elevation Model; DEM), surface temperature, and modeled irradiance over a  $200\,\text{m}\,\text{x}\,200\,\text{m}$  test site on relatively uniform landfast sea ice  $(2.4\pm0.04\,\text{m})$  thick) in McMurdo Sound, Ross Sea, Antarctica. A key technical advance presented here is a new algorithm to correct thermal camera drift from Non-Uniformity Correction (NUC) events in the DJI Matrice 30T, enabling consistent, accurate airborne temperature retrievals with applications beyond polar research. Based on MagnaProbe measurements, the average snow depth for the test site is  $0.1\pm0.04\,\text{m}$ . Snow surface temperatures average  $-14.7\pm0.4\,^{\circ}\text{C}$ , with local variations up to  $12\,^{\circ}\text{C}$ . Small-scale topography strongly affects local irradiance (modeled  $592\pm45\,\text{Wm}^{-2}\,\text{vs}$ .  $593\pm20\,\text{Wm}^{-2}\,\text{measured}$ ), revealing that flat-surface assumptions underestimate local variability of irradiance. Statistical analyses identify irradiance and visible sediment deposition as dominant predictors of surface temperatures, while snow depth plays only a minor role. These results highlight that assuming that snow-covered sea ice is a flat surface fails to represent the full irradiance range, potentially impacting non-linear energy balance processes. Our study provides new insights into drivers of snow surface temperatures over sea ice with potential implications for the sea-ice energy balance.

#### 1 Introduction

The effects of climate change have far-reaching consequences in polar environments, particularly in the ice-dominated high-latitude regions of the Arctic and Antarctica (Callaghan et al., 2011; Simmonds, 2015; Lindsay and Schweiger, 2015; Meredith et al., 2022). These regions experience polar amplification, a phenomenon where warming is more pronounced compared to lower latitudes (Stuecker et al., 2018; Wendisch et al., 2022). As polar ice caps, glaciers, and sea ice melt, the exposed darker surfaces, such as the open ocean and bare land surfaces, can absorb more solar energy, leading to further melting and darkening, and thus influencing large-scale atmospheric circulation patterns (Perovich et al., 2007).

<sup>&</sup>lt;sup>1</sup>Antarctic Research Centre, Victoria University of Wellington, Wellington, New Zealand

<sup>&</sup>lt;sup>2</sup>WSL Institute for Snow and Avalanche Research SLF, Davos, Switzerland

<sup>&</sup>lt;sup>3</sup>Finish Meteorological Institute FMI, Helsinki, Finland

Sea ice growth and decay play a key role in this ice-albedo feedback loop (Riihelä et al., 2021). The annual growth cycle of sea ice is one of the most extensive changes observed on Earth's surface (Parkinson, 2014), and its long-term trend differs significantly between the Arctic and Antarctica. Arctic sea ice has experienced a trend of rapid sea ice loss (Lindsay and Schweiger, 2015). In contrast, the minimum Antarctic sea ice extent exhibits more variability, with a general increase through the satellite period (since 1978) until 2016, and a decrease thereafter. Recent records show the lowest minimum extent since record-keeping began in 1978 (Wang et al., 2022). The thickness, extent, and duration of sea ice impact the absorption and reflectance of solar radiation, thereby influencing the Earth's energy balance (Massom and Stammerjohn, 2010). Additionally, sea ice affects ocean circulation patterns, ecosystems, and the carbon cycle (Stein et al., 2020; Massom and Stammerjohn, 2010).

Snow cover is a critical factor in sea ice thermodynamics, impacting interactions between sea ice and the atmosphere (Landrum and Holland, 2022). Snow, one of the most insulating natural materials, is a barrier to heat exchange between the ocean, sea ice, and atmosphere (Webster et al., 2018). It has a high albedo, up to 0.9, compared to the albedo of bare sea ice (~0.4-0.6) or the open ocean (0.06) (Perovich et al., 2007; Brandt et al., 2011; Light et al., 2022; Smith et al., 2022), enhancing reflectance and influencing regional and global climate by affecting heat and mass balance (Zatko and Warren, 2015). Snow accumulation and metamorphism impact the conductive and radiative properties of sea ice, affecting its thickness, salinity, temperature, and permeability, highlighting the importance of understanding snowpack dynamics (Sturm et al., 1997; Perovich and Polashenski, 2012). Despite the recognized influence of snow properties on sea ice thermodynamics, the relative importance of different factors - such as snow depth, surface microtopography, and sediment deposition - on snow surface temperature remains poorly understood.

Comprehensive datasets concerning the spatial evolution of snow on sea in Antarctica are limited(Eicken et al., 1994; Massom et al., 2001; Brandt et al., 2005; Sturm and Massom, 2016; Arndt and Paul, 2018). Antarctic snow on sea ice differs markedly from its Arctic counterpart, with large spatial variability depending on the location. Antarctic snow is generally thicker (Arndt et al., 2017; Arndt and Paul, 2018; Nicolaus et al., 2021; Lawrence et al., 2024), colder, and persists throughout the year, contrasting with the thinner, warmer snow in the Arctic (Massom et al., 2001). In McMurdo Sound, snow on sea ice persists through summer until breakout, but is generally thinner than in other Antarctic fast ice regions (Brett et al., 2020). This persistent snow cover influences sea ice albedo, primarily through snow thickness and physical properties such as specific surface area, salinity and liquid water content in summer (Zhou et al., 2001). In some areas in Antarctica, like the Weddel Sea, the thick snow cover and thinner sea ice lead to flooding, where the weight of the snow depresses the ice surface, allowing seawater to infiltrate the snow. This seawater-saturated snow then refreezes, resulting in snow-ice formation and thickening of the ice from above (Eicken et al., 1994; Massom et al., 2001; Arndt et al., 2017).

In McMurdo Sound, in particular, research on snow is sparse (Price et al., 2013, 2019; Brett et al., 2020). Recent advancements, such as the Tan et al. (2021) airborne study provide valuable data into snow depth distribution, but underscore the need for more extensive datasets to address critical gaps in our understanding of Antarctic snow on sea ice dynamics. The lack of comprehensive snow data on Antarctica sea ice, and the significant differences between the

Antarctic and Arctic lead to biases in modeling sea ice behavior and errors in global climate system scenarios (Wever et al., 2021; Landrum and Holland, 2022).

To collect data on the physical properties of snow on Antarctic sea ice, drones are a promising tool (Gaffey and Bhardwaj, 2020; Pirazzini et al., 2021; Román et al., 2024). They have been widely used in scientific studies of the cryosphere outside the polar regions (Buhler et al., 2016; Sproles et al., 2020). Drones can cover extensive areas and simultaneously measure specific snow parameters at high resolution, bridging the gap between point-based ground measurements and low-resolution satellite remote sensing (Wigmore et al., 2019; Tan et al., 2021; Wigmore and Molotch, 2023), and even provide opportunities for validating satellite remote sensing products. For example, airborne laser scanning has been used to assess ICESat-2's ability to capture sea ice surface topography and roughness, revealing limitations in detecting small-scale features such as ridges and cracks (Ricker et al., 2023). Similarly, drone-based observations could contribute to validating missions like Polar Radiant Energy in the Far Infrared Experiment (PREFIRE), which aims to improve our understanding of the Earth's energy budget in polar regions (Kahn et al., 2020).

Our study has three objectives: (1) To develop and validate an airborne method for accurately mapping snow surface topography and surface temperature on flat sea ice in the polar regions. (2) To use this airborne method to quantify the snow topography and surface temperature variability over landfast, flat Antarctic sea ice at the 100 m-scale. (3) To investigate the relative influence of potential drivers - including snow depth, sediment presence, and microtopography-driven irradiance variations - on snow surface temperature. We used multi-spectral (visible, thermal infrared) Uncrewed Aerial Vehicle (UAV) imagery at sub-metre spatial resolution, closely following the workflow published by Wigmore et al. (2019). We integrate airborne surface elevation and temperature data collected in November 2022 with ground-based surveys of snow depth, snow surface temperature, and ice thickness. The resulting datasets, combined with topography-dependent irradiance modeling, allowed us to examine how fine-scale surface topography and local sediment deposition control snow surface temperature variability. Our findings demonstrate that no single factor alone can explain the complexity of the sea ice surface energy balance. Our work not only provides new insights into the role of small-scale variability in snow-atmosphere interactions but also delivers a novel, high-resolution dataset to support improved parameterizations of sea ice models.

This paper first describes the study site, ground-based and airborne data, and processing methods, including UAV thermal imaging, Digital Elevation Model (DEM) generation, and the resulting snow depth proxy. The results explore spatial patterns of snow depth and surface temperature, followed by a discussion on technical considerations, and the roles of snow depth, sediment, and irradiance on snow surface temperature.

#### 2 Methods

60

70

During a field campaign in November 2022 (4 weeks), we collected snow and sea ice data on the fast ice in front of the McMurdo Ice Shelf, Ross Sea, about 5 km southwest of Scott Base and McMurdo Station (Fig. 1). We collected a new and unique airborne dataset consisting of aerial Red, Green, Blue (RGB) and Thermal InfraRed (TIR) images of

the snow surface along with photogrammetrically derived digital elevation models, the latter also used to model local irradiance patterns. We calibrated the airborne dataset with ground-based surface temperature, snow depth, and ice thickness surveys, as well as radiation measurements.

**Figure 1.** Landsat image from November 6, 2022. Study area (black star) in West Antarctica, McMurdo Sound, and Ross Island with Scott Base (NZ), McMurdo Station (USA), and the Camp Site (CS) test field (yellow rectangle) on the ice wedge between the Ross Ice Shelf and the tip of Ross Island.

#### 2.1 Study site and conditions

95 McMurdo Sound is located in West Antarctica, in the southwestern part of the Ross Sea, framed by Ross Island to the east, the Ross Ice Shelf to the south, and Victoria Land to the west (Fig. 1). McMurdo Sound has a variable snow cover of 0.1 - 0.3 m (Brett et al., 2020) and limited snow-ice formation compared to wetter regions like the Weddell Sea (Maksym and Markus, 2008). The landfast sea ice is usually 2 - 2.5 m thick (Brett et al., 2020), and there is an extensive sub-ice platelet layer (up to 9 m) due to the proximity of the Ross Ice Shelf and its super-cooled water supply (Gough et al., 2012; Langhorne et al., 2015; Haas et al., 2021). During our field campaign, the average wind speed was 4.7 ms<sup>-1</sup>, and the average temperature was - 14.2 °C (Fig. A2).

Typically, sea ice formation starts in March and lasts until October, followed by a break-out in January/February. In 2022, the sea ice formation was highly impacted by southerly storms repeatedly interrupting the sea ice formation cycle, similar to the conditions in winter 2019 (Leonard et al., 2021). Except for a small "wedge" of ice pinned between the front of the ice shelf and the tip of Ross Island (Fig. 1), the Sound remained largely ice-free until August 2022. As a result, two distinct sea ice types formed: "old ice", which began forming in the typical cycle around March/April and reached a

**Table 1.** Summary of ground-based and airborne instruments with measurement parameters and uncertainties. The uncertainty for the airborne snow depth (\*) is the propagated error counting for the MagnaProbe snow depth error and the GCP error in the z-direction.

| Airborne | <b>Ground-based</b> | Parameter                 | Instrument/Method                  | Unit               | Uncertainty           |
|----------|---------------------|---------------------------|------------------------------------|--------------------|-----------------------|
|          | х                   | Ice Thickness             | Geonics EM31                       | m                  | ± 0.1                 |
|          | х                   | Snow Depth                | SnowHydro MagnaProbe               | m                  | ±0.02                 |
| х        |                     | Snow Depth (Proxy)        | DJI Matrice 30T                    | m                  | ± 0.045 *             |
|          | х                   | Surface Temperature       | Apogee SI-121-SS                   | ℃                  | ±0.2                  |
| Х        |                     | Surface Temperature       | DJI Matrice 30T                    | ℃                  | ±2                    |
|          | х                   | Wind Direction            | R.M. Young Heavy Duty Wind Monitor | 0                  | ±3                    |
|          |                     |                           | HD-Alpine                          |                    |                       |
|          | х                   | Wind Speed                | R.M. Young Heavy Duty Wind Monitor | $\mathrm{ms}^{-1}$ | ± 0.3                 |
|          |                     |                           | HD-Alpine                          | ms                 |                       |
|          | х                   | Air Tomporaturo           | Vaigala LIMD155 Tamparatura Draha  | ℃                  | ± (0.226 - 0.0028     |
|          |                     | Air Temperature           | Vaisala HMP155 Temperature Probe   |                    | × temperature)        |
|          | х                   | Relative Humidity         | Vaisala HMP155 Temperature Probe   | %                  | ±2%                   |
|          | х                   | Incoming (direct/diffuse) | Delta-T Devices                    | ${ m Wm^{-2}}$     | ±5%                   |
|          |                     | Shortwave Radiation       | SPN1 Pyranometer                   | VVIII              | Daily integrals       |
|          | х                   | Incoming/Outgoing         | Kipp & Zonen                       | $\mathrm{Wm^{-2}}$ | $\pm$ 1.2 % to        |
|          |                     | Shortwave Radiation       | CMP22 Pyranometer                  | VVIII              | $\pm$ 1.4 %           |
|          | х                   | Incoming/Outgoing         | Kipp & Zonen                       | $\mathrm{Wm^{-2}}$ | $\pm$ 1.5 % to        |
|          |                     | Longwave Radiation        | CGR4 Pyrgeometer                   | VVIII              | $\pm2.0\%$            |
|          | V                   | Location (Long/Lat)       | Septentrio AsteRx                  | m                  | $\pm$ 0.01 horizonta  |
|          | X                   | Location (Long/Lat)       | Septentino Asterix                 | m                  | $\pm$ 0.02 vertically |

thickness of about 2.5 m during our field campaign, and "new ice", which started forming only after August 2022 and grew to about 1 - 1.5 m thickness by November 2022. During the November 2022 field campaign, we collected data on five 200 m x 200 m sites with different snow and ice conditions, of which, in this study, we only discuss the site we assumed to be "flat and relatively homogeneous". The site was our main test site (Camp Site; CS), located on the flat "old ice", next to the field camp (Fig. 1, yellow rectangle).

#### 2.2 Ground-based data

Our ground-based dataset consists of automated radiation data, weather data, and snow and ice thickness measurements. All sensors, instruments, and parameters used in this study are listed in Table 1.

#### 115 2.2.1 Radiation and weather data

We installed automated measurement stations to capture the local radiation, surface temperature, and weather conditions and to validate and correct airborne measurements, as well as to calibrate the irradiance model. The radiation station was outfitted with a pair of Kipp & Zonen CMP22 pyranometers, one facing upward and one facing downward, to measure the broadband incident (Sw $\Downarrow$ ) and reflected (Sw $\Uparrow$ ) solar irradiance in the 200 - 3600 nm wavelength range, and a pair of Kipp & Zonen CGR4 pyrgeometers, one facing upward and one facing downward, to measure the broadband incoming (Lw↓) and outgoing (Lw↑) longwave radiation in the 4500 - 42000 nm wavelength range. The sensors were ventilated, and according to the manufacturer's specifications, the nominal accuracy of the Sw and Lw measurements is ~1% and 3%. respectively. The radiation station also included a DeltaT SPN1 radiometer that measured broadband global and diffuse incident irradiance (Sw and Sw<sub>diff</sub> ↓) in the 400 - 2700 nm wavelength range. According to the manufacturer, the nominal accuracy of both measurements is  $\pm$  8%. Inter-calibration studies have shown that an extra factor of 1.05 should be applied to Swdiff 11 to compensate for its systematic underestimation (Badosa et al., 2014). A stationary Apogee infrared radiometer (model SI-121-SS) installed slightly north of the radiation station was pointing at clean snow ("cold target") with an angle of approximately 45 ° mounted at ~1.2 m height, leading to an elliptical footprint of about 1.6 m<sup>2</sup>. The data were logged every 10 s. A non-stationary Apogee infrared radiometer (model SI-121-SS) on a tripod was placed in the test field and mounted at 1.02 m, pointing vertically downward (nadir view) a sediment patch ("sediment target"), yielding a near circular footprint of 0.35 m<sup>2</sup>.

## 2.2.2 Snow depth

To capture the spatial distribution of snow depth, we conducted MagnaProbe transects (Sturm and Holmgren, 2018). The MagnaProbe is a user-friendly device that allows for rapid snow depth measurements. The snow depth is measured by inserting a 1.53 m steel rod with a sliding ring magnet inside a plastic disk into the snow until the tip of the rod hits the ground. An electronic pulse from a magnetostrictive sensor determines the position of the magnet, converting the signal into the distance from the rod's tip to the disk and indicating snow depth. The snow depth and the GPS position (GARMIN $^{\text{TM}}$  receiver and Campbell Scientific antenna) are recorded with a Campbell Scientific CR800 data logger (Sturm and Holmgren, 2018). The accuracy of the snow depth depends on (1) the hardness of the underlying surface (a too-soft material will cause compaction and over-probing), (2) the positioning of the plastic disk on the snow surface, and (3) the correctness of the vertical angle. For our measurement campaign, the uncertainty is mainly influenced by (2) and (3) as the snow surface is relatively hard and uneven, affecting the insertion of the probe and the ability of the plastic disk to lie flat on the snow surface. We assume the measurements are within a  $\pm$  0.02 m uncertainty. This is slightly larger than the 0.01 m assumed for the winter MOSAiC measurements over likely similar roughness (Itkin et al., 2023), to account for the exceptionally hard snow surface, which made it difficult for the MagnaProbe disc to lay perfectly flat. The GPS accuracy is  $\pm$  2.5 m. Our measurement strategy was to conduct transects along the measurement field borders and a

cross through the middle, taking a measurement approximately every 1.5 m, which resulted in 813 measurements for this particular "study field".

# 2.2.3 Ice thickness

We used the Geonics electromagnetic conductivity meter (EM31) combined with manual drilling to survey the ice thickness. The EM31 is a geophysical instrument that measures sea ice conductivity and in-phase components to detect subsurface features (Tateyama et al., 2006). It operates based on the principles of electromagnetic induction. The EM31, consisting of a transmitter and receiver coil, was mounted on a sledge and towed over the snow by a snow machine. The transmitter coil emits a primary electromagnetic field into the ground, and as the field penetrates the ice, it reaches the conductive seawater beneath. This causes strong eddy currents to form, generating a secondary electromagnetic field. The receiver coil detects this secondary field, which is altered by the distance it travels through the ice. The EM31 measures the apparent conductivity of the subsurface and the phase shift between the primary and secondary fields. These measurements are recorded continuously as the instrument is moved across the ice. To account for local variations in ice thickness and device calibration, we took direct ice thickness measurements at various points (approximately every 50 m) using a Covacs ice drill. We used the same survey pattern for the EM31 ice thickness measurements as for snow depth. While there is a significant platelet ice layer in our survey field, we excluded it from our ice thickness evaluation by calibrating the EM31-signal to ice and snow thickness only. The snow and ice thickness was calculated following a linear regression between conductivity and manually measured ice and snow thickness at the CS test field:

$$h_{\text{ice+snow}} = -0.0103 \,\kappa + 4.2285$$
 (1)

with  $h_{ice+snow}$  the combined ice and snow thickness and  $\kappa$  the conductivity. During our surveys the EM31 was mounted 0.5 m above the surface (snow or ice depending on the measurement location) on a sledge that was pulled with a snow machine. To calculate only the ice thickness, we subtracted the mounting height as well as the average snow depth of 0.16 m from the DEM.

#### 2.3 Airborne data


We used a DJI Matrice 30T, a multi-rotor (quadcopter) UAV with self-heating batteries, suitable for operations in polar conditions (temperature range from -20 to +50 °C). The UAV has a wide-angle camera (12 megapixels) for RGB images and a thermal camera (Uncooled Vox Microbolometer, long-wave infrared spectrum 8-14  $\mu$ m, 1.31 megapixels) for TIR images with an accuracy of  $\pm 2$  °C or  $\pm 2$ % (Da Jiang Innvoations (DJI), 2022). The exact wavelength for the wide angle and TIR camera is unknown.

To retrieve RGB and TIR imagery, we programmed two missions following a lawnmower cross-grid survey pattern at two different flight altitudes, and executed them on November 14, 2022. The first flight at 60 m altitude took off at 22:12:00 and finished at 22:31:19 (UTC). We changed the batteries and executed the second flight at 75 m altitude, with the take-off at 22:44:45 and landing at 23:01:37 (UTC). The frontal overlap of the images was set to the maximum of 95% and

the side overlap to 80 %. The high frontal overlap only affects the frequency at which the pictures are taken and does not affect the flight pattern or flight time. The RGB and TIR images were shot simultaneously.

# 2.3.1 Georeferencing







To georeference the RGB and TIR images, we used 10 targets as Ground Control Points (GCPs) on the snow surface across the 200 m x 200 m measurement area before the drone flight. The RGB targets were 0.3 x 0.3 m sheets of yellow coroplast plastic, with a pink duct-tape cross defining a centre point. For the TIR survey, 0.4 x 0.4 m black targets were installed next to the RGB targets. As black (hot) targets, we used thermal insulation material, similar to camping mats, that was black on the top side, with silver-colored foil as backing (similar to first aid blankets) at the bottom. The location of each of the RGB targets was measured using dual-frequency GNSS receivers, and post-processing was done using the Precise Point Positioning kinematic method (Malinowski and Kwiecień, 2016) and the software «teqc» (Estey and Meertens, 1999).

## 190 2.3.2 TIR image processing: Non-Uniformity Correction (NUC)

The proprietary RJPG format used by the thermal camera retains absolute brightness temperatures but cannot be directly analyzed. We therefore converted the images to TIFF using the ImageJ IR UAV plugin (Pereyra Irujo, 2022) before processing them with the SfM workflow in Agisoft Metashape Pro version 1.8.4 (Wigmore and Molotch, 2023) in Fig. 2.

The thermal camera of the DJI Matrice 30T uses an uncooled microbolometer, which is prone to non-uniformities in its sensor array; each pixel responds slightly differently to the same temperature. To correct for measurement drift caused by these differences, the camera performs periodic internal calibrations known as Non-Uniformity Correction (NUC) events. This process involves a shutter that blocks the lens, providing a uniform temperature source for calibration. The camera measures each pixel's response to this source and adjusts its settings to correct any discrepancies. After calibration, these adjustments ensure accuracy and uniformity in subsequent images. The frequency of NUC events varies depending on the camera model and environmental conditions, but it is often not documented and seldom adjustable for off-the-shelf drones, like the DJI Matrice 30T. Consequently, temperature differences of up to 1.5 °C can occur in consecutive TIR images taken before and after a NUC event, as shown in Fig. 3a (red and yellow dots, respectively). We introduce a novel NUC correction algorithm that aligns and compares temperature measurements across frames, mitigating discontinuities caused by internal NUC events:

Step 1: Read sequential image pairs (RJPG; 1280 x 1024 px)

Step 2: Identify overlapping areas in each image pair (in this dataset, the overlap is at least 70%). We do this by finding and extracting unique features to ensure identical regions in each image pair. For this, we use the Oriented FAST and Rotated BRIEF (ORB algorithm) detector, which combines two key components: FAST (Features from Accelerated Segment Test) and BRIEF (Binary Robust Independent Elementary Features Descriptor). FAST identifies interesting points (corners) in an image, similar to spotting unique landmarks on a city map. BRIEF then describes

**Figure 2.** Workflow for processing and analyzing RGB and TIR images captured with the DJI Matrice 30T during two flights in 60 and 75 m altitude. The process includes converting RJPG images to TIFF format using the ImageJ IR UAV plugin, applying the NUC correction algorithms for the internal camera calibration, and processing both TIR and RGB images through photogrammetry steps in Metashape. Key steps include sparse point-cloud generation, georeferencing, optimization, dense cloud generation, mesh creation and the DEM and RGB/TIR orthomosaic production. Then, we apply a final offset correction using ground-based MagnaProbe snow depth measurements and the surface temperature calibration using ground-based Apogee temperature data.

these corners using compact binary "fingerprints," analogous to noting a landmark's distinctive features (e.g., "a red brick house with a tall chimney"). The ORB algorithm enables efficient and robust feature matching between images (Rublee et al., 2011; Bradski, 2000). Then, we use the Brute-Force Matcher (Bradski, 2000) to calculate corresponding points and select the 50 best feature matches in each image pair to estimate a homography matrix, which describes the geometric relationship between the two images.

**Step 3:** For each pair of successive images, the first image is warped to align with the grid of the second image to ensure accurate overlap and compensate for camera movement.



- **Step 4:** Extraction of brightness temperatures from each pixel. From here on, we use the converted TIFF images (TIFF; 640 x 512 px).
- Step 5: For each TIFF image, calculation of the mean brightness temperature  $\overline{T}_{brightness}$  of the area overlapping with the following image. Ideally, the  $\overline{T}_{brightness}$  of the same overlapping area in different images should match, as the thermal images were captured nearly simultaneously. However, due to temperature drift and NUC events, discrepancies arise and must be corrected.
- Step 6: Identification of NUC events. A NUC<sub>i</sub> event, with i=1,...m (m being the total number of NUC events during the considered flight), is identified when the absolute differences between the  $\overline{T}_{brightness}$  of overlap regions in consecutive images is  $> 0.5 \,^{\circ}$ C (Fig. 3a). NUC events in this study were identified using a fixed threshold of  $0.5 \,^{\circ}$ C, chosen empirically based on the stability of the correction curve (Fig. 3a). We tested different thresholds ( $1 \,^{\circ}$ C,  $0.5 \,^{\circ}$ C, and a dynamic 99th percentile approach) and found that  $0.5 \,^{\circ}$ C visually identified the NUC events most clearly and consistently at this site. We therefore chose this fixed threshold for the present analysis, while noting that adaptive approaches may be more appropriate at sites with greater temperature heterogeneity. Between two NUC events, NUC<sub>i</sub> and NUC<sub>i+1</sub>, there are n images  $I_j$ , with j=1,...n (n being the number of TIR images taken between two consecutive NUC events). For image  $I_j$ ,  $\overline{T}_{brightness(i,j)}$  represents the mean temperature of image  $I_{j+1}$  (in the overlap region). We do not consider a temperature change  $> \pm 0.5 \,^{\circ}$ C to be a NUC event if it affects one image only (e.g. Fig. 3a: blue dot).
- Step 7: Correction of the temperature drift between two consecutive NUC events. We assume that the temperatures measured immediately after the NUC events are accurate, and that the temperature drift (drift rate d<sub>i</sub>) over the n images between two consecutive NUC events (NUC<sub>i</sub> and NUC<sub>i+1</sub>) is linear:

$$d_{i} = (\overline{T}_{brightness(i+1,1)} - \overline{T}_{brightness(i,n)})/n$$
(2)

where  $\overline{T}_{brightness(i,n)}$  is the mean temperature of the overlap area in the final image n in the series after the NUC<sub>i</sub> event (red dots in Fig. 3), and  $\overline{T}_{brightness(i+1,1)}$  is the mean temperature of the overlap area in the first image after the

 $NUC_{i+1}$  event (yellow dots in Fig. 3).  $d_i$  is then applied to calculate the linear correction  $c_{i,j}$  for the  $\overline{T}_{birghtness(i,j)}$  mean temperatures of the j=1,...n images between  $NUC_i$  and  $NUC_{i+1}$ :

$$c_{i,j} = j d_i \tag{3}$$

The brightness temperature of each pixel in each TIR image is then corrected by adding ci.i.

To understand the improvement achieved with this temperature correction procedure, we analysed the temperature in 250 random areas of  $10 \times 10 \, \text{m}^2$  taken from the NUC uncorrected and NUC corrected TIR orthomosaics. While our algorithm to correct the images for NUC events is applied sequentially (it depends on the order of images), we found that this analysis does not depend on the order in which the images were taken. For each of the 250 areas that contained at least 10 images, we calculated the temperature anomaly per image by subtracting the mean areal temperature from the mean image temperature  $\overline{T}_{brightness}$ . By comparing the temperature anomalies in the NUC uncorrected and NUC corrected TIR orthomosaic, we can assess the algorithm's impact performance in reducing drift and variability of temperature anomalies (Fig. 3c, dark grey vs. light grey) across the dataset.

In the NUC uncorrected TIR orthomosaic, the spread of anomalies ranged between  $\pm 4$  °C due to the temperature drift in between NUC events and the consequent variability in  $\overline{T}_{brightness}$  among the images capturing the same overlapping area. In the NUC corrected TIR orthomosaic, the distribution of temperature anomalies narrowed to the range  $\pm 2$  °C, there were fewer outliers, and the whiskers of the temperature anomaly boxplot shrank (Fig. 3c). This suggests that our correction improved the repeatability of  $\overline{T}_{brightness}$  at each site, reaching an accuracy comparable to the sensor uncertainty (see Section 2.3). The Root Mean Square (RMSE) of the temperature anomalies across all 250 sampled areas was 1.11 °C in the NUC uncorrected orthomosaic and decreased by 0.53 °C down to 0.58 °C in the NUC corrected orthomosaic.

#### 2.3.3 TIR orthomosaic





To produce the TIR orthomosaic, we first generated the sparse point cloud through tie-point generation, alignment, and cloud creation. After georeferencing with GCPs from both RGB and TIR images, we optimized the sparse cloud by removing outliers. We then created a high-quality, dense point cloud, generated a TIN mesh, and applied an oval-shaped vignette correction mask. The vignette correction mask is applied to remove the areas affected by the temperature distortion at the edges (corners) of the TIR images, which tend to be cooler than the centre due to lens properties, sensor characteristics, and optical aberrations (Wigmore and Molotch, 2023). We then calculated the first orthomosaic, we manually corrected motion blur in the corner areas of the DEM by excluding these images, and produced the final orthomosaic, which was then calibrated using the ground-based Apogee temperature dataset (see the following Section 2.3.4). The TIR orthomosaic has a reprojection error of 0.6 px and GCP errors in x, y and z-directions of 0.008 m, 0.003 m and 0.001 m. The workflow is summarized in Fig. 2.

Figure 3. Illustration of NUC events (a) and their correction (b) for the first flights at 60 m altitude. (a) Mean brightness temperature ( $\overline{T}_{brightness}$ ) for overlap area per image. Red dots indicate  $\overline{T}_{brightness}$  before the NUC event, and yellow dots indicate  $\overline{T}_{brightness}$  after the NUC event. NUC events are temperature jumps of more than 0.5 °C in two consecutive images. The blue dot indicates a temperature outlier for one image, which is not a NUC event. (b) The same dataset after the correction algorithm (Eq. 2 and Eq. 3) is applied. (c) The Kernel Density Estimate (KDE) function showing the distribution of brightness temperature anomalies in the 250 randomly sampled areas of 10 x 10 m<sup>2</sup> taken from the NUC uncorrected (dark grey) and NUC corrected (light grey) TIR orthomosaics.

#### 2.3.4 TIR orthomosaic: Ground-based temperature calibration



To derive the surface temperature orthomosaic, we calibrated the brightness temperatures of the TIR orthomosaic using surface temperature data obtained from sea-ice-based observations. We calculated the snow surface temperature from the brightness temperature measured by the Apogee infrared radiometer, accounting for the contribution of the reflected fraction of downwelling longwave radiation measured with a Kipp & Zonen CGR4 pyrgeometer (Apogee Instruments INC., 2022):

$$T_{\text{surface}} = \sqrt[4]{\frac{T_{\text{Apogee}}^4 - (1 - \epsilon)Lw_{\downarrow}/\sigma}{\epsilon}}$$
 (4)

where  $T_{surface}$  is the surface temperature (K),  $T_{Apogee}$  is the surface brightness temperature measured with the Apogee sensor (K),  $\epsilon = 0.98$  is the snow thermal emissivity, and  $\sigma$  is the Stephan-Boltzman constant.

In Fig. 4a, we present the surface temperature time series of the snow surfaces within the footprints of the two Apogee infrared radiometers during the flights, alongside the  $T_{brightness}$  of the corresponding TIR images that cover the two footprints (with a RMSE of 0.58 °C quantifying the residual error in the temperature dataset and serving as an uncertainty estimate for the surface temperature maps). We extracted  $T_{brightness}$  within the Apogee sensor footprints from the TIR

Figure 4. (a) The time series (line plots) of surface temperatures measured on November 14th, 2022 with the two Apogee sensors (Table 1) targeting snow with visible sediment particles (yellow), and clean snow without visible sediment particles (blue). The dots are  $T_{brightness}$  from all TIR images (NUC corrected) that contain the Apogee sensor's footprint with the temperature anomaly RMSE of 0.58 °C that results from the correction equation (eq. 2 and 3). (b) Scatter plot with ground-based temperatures measured by Apogee sensors ( $T_{surface}$ ) at the two targets against the  $T_{brightness}$  from TIR images that encompass the footprints of these sensors. The black line is the best linear fit. The error bars indicate the temperature anomaly RMSE of 0.58 °C and the manufacturer's uncertainty for the Apogee sensor, respectively.

images and plotted it against the Apogee-measured surface temperatures (T<sub>surface</sub>) in Fig. 4b. We calibrated the airborne brightness temperatures to surface temperatures by calculating the best (linear) fit between the Apogee-measured, sea ice-based surface temperatures T<sub>surface</sub> and the collocated, simultaneous TIR image brightness temperatures T<sub>brightness</sub>. To match the airborne data points with the temporally higher resolved (10 s) sea ice-based measurements, we averaged the three T<sub>surface</sub> measurements taken before and after the T<sub>brightness</sub> timestamp. We used this dataset to calculate the linear fit:

$$T_{calibrated} = 0.68 T_{brightness} - 4.44$$
 (5)

This equation is applied to each pixel in the TIR orthomosaic. The total uncertainty of the derived surface temperature is calculated as the square root of the sum of the following squared contributions: the RMSE of the linear fit (0.48 °C), the thermal camera uncertainty ( $\pm$ 2 °C), and the RMSE associated with the NUC correction (0.58 °C). This results in a total uncertainty of  $\pm$ 2.1 °C for the surface temperature derived from airborne thermal imagery.

#### 2.3.5 Digital Elevation Model (DEM)


We derived the DEM from the RGB images following the workflow detailed in Fig. 2. First, we generated the sparse point cloud through image alignment and tie-point generation. After georeferencing using GCPs from both RGB and TIR

images, we optimized the sparse cloud by removing outliers with poor accuracy values. We then created a dense point cloud, generated a triangular irregular network (TIN) mesh and finally calculated the RGB orthomosaic (2 cm/px ground resolution) and DEM (5 cm/px ground resolution). We upscaled both to 9 cm/px to match the TIR orthomosaic resolution. The projection error for the RGB orthomosaic and DEM is 0.96 px, and the GCP errors in x, y, and z directions are 0.05 m, 0.05 m and 0.04 m.

As the last processing step, we detrended the DEM by fitting a planar surface to the elevation data and subtracting this fitted plane from the original DEM. The trend is likely due to the time window of about 3 hours between when the GCP survey was conducted and when the site was flown, as the site is influenced by tidal variations, which are about 0.2 m for this 3 - hour time window as shown in Fig. A1. We fitted a plane through the DEM to remove this trend, determined the best-fit parameters for the plane, and subtracted it from the original DEM, producing a detrended DEM. This allowed us to focus on analyzing the smaller-scale topographical features of the snow surface, which is the aim of this study.

## 2.3.6 Snow depth proxy








Mapping snow depth typically requires surveying the area both when it is snow-free and when it is snow-covered. However, this approach was not feasible in our case because the underlying surface is sea ice, which was never snow-free (for our field work window) and continuously changed in elevation due to tides, thickening, and thinning processes. To overcome this limitation and generate a continuous snow depth estimate, we derived a snow depth proxy from the detrended UAV-based DEM, under the assumption that the underlying sea ice surface was smooth and nearly flat (Fig. A3). Because the DEM and snow thickness share the same vertical scale, a relative change in DEM elevation (e.g., 0.10 m) corresponds directly to an equivalent change in snow thickness measured with the MagnaProbe. Thus, even though the absolute offset is unknown, the proxy retains the correct variability, which is sufficient for our analyses. The main limitation is that sea ice is not perfectly flat, which introduces some additional uncertainty.

Instead of spatially aligning individual MagnaProbe measurements to the DEM, we used a single value offset between the lowest DEM elevation and the shallowest measured snow depth to correct the DEM surface. Specifically, the minimum snow depth measured with the MagnaProbe  $(0.012\,\text{m})$  and the minimum DEM elevation  $(-0.149\,\text{m})$  differ by  $0.16\,\text{m}$  as shown in Fig. 5a. We therefore added a uniform  $0.16\,\text{m}$  offset to the entire DEM to approximate snow depth for each pixel as illustrated in Fig. 5b. Differences between the distributions are expected because the DEM and MagnaProbe surveys capture snow depth at different spatial scales and sampling densities. The MagnaProbe transect data  $(\pm\,0.02\,\text{m})$  snow depth error) provided the reference for this offset but could not be used for a direct geospatial match because of the limited accuracy of the MagnaProbe GNSS positions. As a result, the corrected DEM values should be interpreted as a proxy for snow depth rather than an exact measurement. The propagated vertical uncertainty (per-pixel vertical measurement uncertainty), which combines the DEM vertical error  $(\pm\,0.04\,\text{m})$  from GCPs) and the MagnaProbe error, is estimated at  $\pm\,0.045\,\text{m}$ . Since we do not co-locate DEM pixels to exact MagnaProbe points for point-wise validation, we are not propagating a horizontal offset through a point-by-point comparison.

**Figure 5.** (a) Histogram showing the estimated Probability Density Function (PDF) of elevation (m) from the DEM and snow depth (m) from the MagnaProbe transect. The minimum elevation is offset by 0.16 m relative to the minimum snow depth. (b) Histogram (PDF) of the offset-corrected DEM elevation (m), used as a proxy for snow depth, and the measured snow depth (m) from the MagnaProbe transect.

This approach, while simplified, is supported by field observations indicating smooth, flat ice surface conditions across the study site as shown in Fig. A3. However, it does not represent an absolute snow depth, or capture any localized sea-ice surface roughness that was not sampled, which should be considered when interpreting the proxy snow depths.

The test field was situated on sea ice with an average thickness of  $2.4 \pm 0.04 \, \mathrm{m}$  across 2449 measurements. The relative variability in sea ice thickness was calculated as the mean absolute deviation of all measurements expressed as a percentage of the mean thickness, yielding 1 % (Fig. 6). Thickness ranged from  $2.28 \, \mathrm{m}$  to  $2.52 \, \mathrm{m}$  with a measurement uncertainty of  $0.1 \, \mathrm{m}$  (Table 1). Since sea ice is buoyant, changes in thickness lead to only minor surface elevation variations, as most of the ice mass remains submerged. In contrast, snow depth varied significantly more at 32 % (see Table 2), even on this "flat" field, highlighting the snow cover's heterogeneity driven by wind processes.


**Figure 6.** Sea ice thickness (m) distribution for 2449 measurements points surveyed with the EM31. The bandwidth parameter for the KDE function is 2.

**Table 2.** Mean thickness and Standard Deviation (STD) for the sea ice thickness (m), snow depth proxy (m) derived from the DEM and snow depth (m) measured with the MagnaProbe, the number of data points (n) and the relative change (%) for each dataset.

| Thickness    | n       | Mean<br>± STD<br>(m)              | Rel.<br>Change<br>(%) |
|--------------|---------|-----------------------------------|-----------------------|
| Sea ice      | 2449    | $2.4 \pm 0.04$                    | 1                     |
| Snow (DEM)   | 6211943 | $\textbf{0.16} \pm \textbf{0.06}$ | 30                    |
| Snow (Magna) | 813     | $\textbf{0.1} \pm \textbf{0.04}$  | 32                    |

To show that a direct correlation of the DEM elevation values and MagnaProbe measurements is not feasible, we extracted DEM values within a 2.5 m radius around 6 randomly selected MagnaProbe measurement locations to mimic the effect of positioning uncertainty (Tab. 1). For each point, we plotted histograms of the DEM-derived snow depths alongside the measured MagnaProbe depth (Fig. A4).

# 345 2.3.7 Irradiance calculation



We calculated surface irradiance on each DEM grid cell by combining UAV-derived topography (slope and aspect) with solar position at the time of the flight. We based the irradiance calculation on the SPN1 values because they allow partitioning into direct and diffuse components, which is required to apply the geometric slope correction. The radiation values are scaled to match the full-spectrum CMP22 pyranometer This approach follows established methods such as those described by Dozier and Frew (1989). We computed the direct irradiance on a sloped snow surface using the incidence angle between the Sun and local surface normal:

$$\cos\Theta = \cos\beta \cos\theta_z + \sin\beta \sin\theta_z \cos(\phi_s - \phi) \tag{6}$$

with  $\Theta$  the angle of incidence (°),  $\beta$  the surface slope (°),  $\phi$  the surface aspect (°),  $\theta_z$  the solar zenith angle (°), and  $\phi_s$  the solar azimuth angle (°) at the time of the UAV flight. Self-shaded cells are assigned a value of  $\cos \Theta = 0$  and received zero direct radiation.

We scaled the SPN1 radiometer measurements (total  $I_t = 543 \, \text{Wm}^{-2}$  and diffuse  $I_{diff} = 62 \, \text{Wm}^{-2}$  at the time of flight) using the full-spectrum pyranometer ( $I_{SW} = 593 \, \text{Wm}^{-2}$ ) to account for spectral bias using  $\kappa = I_{SW} \, / \, I_t$ , then separated total and diffuse components, correcting the diffuse radiation with a factor 1.05 following Badosa et al. (2014):

$$I_{dir} = \kappa (I_t - 1.05 I_{diff}) \tag{7}$$

where I<sub>dir</sub> is the direct beam irradiance corrected for spectral bias. The slope- and shading-corrected surface irradiance for each DEM grid cell was then calculated as:

$$I = I_{dir} M \cos \Theta + 1.05 \kappa I_{diff}$$
 (8)

where Θ is the incidence angle between the Sun and the local surface normal, and M is the shade mask (0 for shaded cells, 1 for sunlit cells) following Corripio (2003). We note that multiple scattering between snow facets can enhance irradiance, particularly in steep and complex terrain (Robledano et al., 2022). However, given the relatively shallow slopes at our site, we assume this effect is minor and do not account for it in our model.

# 3 Results and Discussion





# 3.1 Airborne maps: red band orthomosaic, snow depth proxy and surface temperature

Our airborne products are shown in Fig. 7. The upper row contains the orthomosaics for red band values, the snow depth proxy (DEM) and surface temperatures. The red band orthomosaic (Fig. 7a) provides an overview of the study area and allows us to identify areas with visible sediment deposition as they are visible by the eye as darker regions (Fig. 7d). We use the red band (wavelengths  $> 0.6 \,\mu m$ ) of the RGB orthomosaic to detect visible sediment deposition on snow. Snow albedo is most sensitive to impurities in the visible wavelengths (Warren and Wiscombe, 1980), and we found the most contrast in the red band, where, qualitatively, sediment patches stand out more clearly against the bright snow background. The lower panel shows a close-up of a region with site-specific surface features. In the close-up two main features stand out: the sediment deposition in the upper left corner and the snow dunes in the lower part. The sunexposed sites of of the snow dunes look "brighter", which could be due to snow thickness or due to irradiance and shading effects. Both features, the sediment and the dunes, are visually distinguishable and provide information for understanding the drivers of surface temperature.

Fig. 7b shows the DEM. The snow depth proxy derived from this DEM ranges from 0.01 to 0.65 m, with a mean of 0.16 and a standard deviation (STD) of  $\pm$  0.06 m (Fig. 8a). We identified values exceeding 0.4 m as snow drifts around obstacles, such as measurement stations, solar panels, and flags based on Fig. 8e and Fig. 8i. The DEM reveals relatively high variability in the snow depth proxy values, mainly due to wind-induced snow dunes (e.g. Fig. 7e, lower left). Although we expected dunes to align with the predominantly southerly winds during our campaign (Fig. A2b), the local aspects of snow slopes (i.e., dune slopes, Fig. 9c) showed no clear preferential orientation. This random distribution suggests that prevailing winds had limited influence on dune alignment during the measurement period. While dune orientation is outside the scope of this study, evaluating a longer period of wind data preceding our field campaign would be necessary to understand dune alignment.

In Fig. 7c, we present the TIR orthomosaic with surface temperatures ranging from - 17.5 to -5 °C. The mean temperature is -14.7  $\pm$  0.4 °C (Fig. 8b). Temperatures exceeding -4 °C are caused by the measurement infrastructure and are masked in the TIR orthomosaic. The sediment patches identified in Fig. 7d are warmer than areas with cleaner snow as sediment has a lower albedo and absorbs more shortwave radiation (Fig. 7f). We also observe that the shaded sides of the snow dunes have lower temperatures (Fig. 7f, lower left) and are also darker in red band orthomosaic (Fig. 7d).

# 3.2 Possible drivers of small-scale surface temperature variations: snow depth, sediment deposition or irradiance?

In this section, we use the UAV-derived maps of snow depth, surface temperature, and red bad values, as well as the modeled map of solar irradiance, to try and disentangle the relevant drivers of local surface temperature variations. All statistical correlations are shown in Table A1.

Figure 7. Airborne products: Red band orthomosaic, snow depth proxy (DEM) and surface temperature orthomosaic of the CS test field) with 9 cm/px resolution. White spots are masked stations and targets (see Fig. A5). The yellow arrow indicates the Sun's position in the north-east. (a) Red band (0 to 255) orthomosaic, with dark patches showing areas with sediment deposition. (b) DEM of the CS showing the snow depth proxy (m). The color bar is set from 0 to 0.6 m to minimize the impact of remaining measurement stations and flags to better show the spatial variability of the snow cover. The minimum snow depth proxy is 0.01 m as shown in Fig. 8a, hence there is now bare ice present at this site. (c) TIR orthomosaic with surface temperature in ℃. The color bar is set from -8 to -16 ℃ to minimize the impact of remaining measurement stations, targets and flags and display the spatial variability of the snow surface temperature. (d) Close-up of the white framed area in (a) with red band values. (e) Close-up of the snow depth proxy in the same area as (d). (f) Close-up of the surface temperature in the same area as (d) and (e).

#### 3.2.1 The role of snow depth




To test if snow surface temperatures are driven by snow depth (given relatively uniform sea ice thickness), we first explore the correlation between the snow depth proxy and snow surface temperature. Given the visible sediment patches and their strong effect on albedo (Warren and Wiscombe, 1980), we split the data into three sub-datasets: 1) the "entire field" dataset, containing 6,211,943 data points (approximately 99.9% of the original dataset), 2) the "sediment" dataset, which includes 285,396 data points (approximately 5%) from areas with visible sediment and some clean snow, and 3) the "no sediment" dataset, containing 341,473 data points (approximately 6%) representing areas with visible clean snow only.

We delineated sediment patches by manually drawing polygonal areas around visible sediment deposits with the QGIS Geographic Information System (Fig. A5). We tested various automatic RGB thresholds to identify impurities, but the results were unsatisfactory, partly because distinguishing between impurities and shaded areas proved challenging. For the "no sediment" dataset, we randomly selected rectangular patches where no visible sediment was apparent, ensuring a similar sized area for comparability.

The scatter plot in Fig. 10a reveals a very weak but statistically significant correlation between surface temperature and snow depth proxy (Spearman correlation coefficient  $r_s$  = 0.16, Table A1) for the "entire field" data set. The mean snow surface temperature and snow depth proxy are -14.7  $\pm$  0.4 °C and 0.16  $\pm$  0.06 m, respectively (Fig. 8b, a). In the "sediment" dataset the correlation is slightly stronger ( $r_s$  = 0.26, Fig. 10d). This suggests that sediment deposition is influenced by the same wind processes as snow depth, with both snow and impurities accumulating in the same areas. The "sediment" dataset has a higher mean surface temperature and greater variability (-13.97  $\pm$  0.82 °C; Fig. 8f). The larger temperature variability shows that the delineated sediment patches include both clean and "dirty" snow, likely reflecting variations in sediment concentration (Fig. 8f). The "sediment" dataset also shows a higher mean snow depth proxy (0.21  $\pm$  0.07 m, Fig. 8e).

The "no sediment" dataset shows the weakest positive correlation between surface temperature and snow depth proxy  $(r_s = 0.06, Fig. 10g)$ . It also has the lowest mean temperature  $(-14.84 \pm 0.28 \, ^{\circ}\text{C}; Fig. 8j)$  – likely caused by the absence of the (visible) warmer sediment patches – and the lowest mean snow depth proxy  $(0.14 \pm 0.05 \, \text{m}, Fig. 8i)$ .

In summary, we conclude that the positive correlation between snow depth and snow surface temperatures is primarily driven by sediment deposition rather than snow depth itself. This emphasizes the role of sediment on albedo and radiation absorption rather than the role of snow depth on thermal resistance. The sediment will have the effect that, when air temperatures rise later in the season, the increased solar radiation absorption by the sediment will reduce the snow specific surface area and accelerate snow melt, further decreasing the albedo and triggering the albedo feedback (Ledley and Pfirman, 1997; Skiles et al., 2018). While snow depth plays an important role in the energy balance of sea ice through albedo and thermal conductivity effects (Warren and Wiscombe, 1980), our data show that it is not the primary driver of surface temperatures in our case, even when excluding visible surface impurities.

**Figure 8.** Histograms with PDFs for the snow depth proxy, surface temperature, red band values and irradiance for the three subdatasets. The first row is the "entire field" dataset, the second row is the "sediment" dataset, and the last row is the "no sediment" dataset. The snow depth bin size (a, e and i) is 2 cm. The temperature bin size (b, f and j) is 0.5 ℃. The red band bin size (c,g and k) is 5. The irradiance bin size (d, h, and l) is 10 Wm<sup>-2</sup>.

#### 3.2.2 The role of "darker" areas: sediment or shadows?

Building on our finding that sediment strongly influences surface temperatures, we correlated the surface temperature with the red band values from the orthomosaic. Sediment particles – like dust, soot, or organic matter – absorb more

sunlight in the visible spectrum than clean snow (Tuzet et al., 2019; Cui et al., 2021). These surface impurities largely influence the snow albedo and, consequently, surface temperatures (Warren and Wiscombe, 1980; Doherty et al., 2010; Réveillet et al., 2022).

We use the red band orthomosaic as a proxy for "surface sediment darkness". While red band intensity is not a perfect quantitative indicator of sediment concentration – and is also influenced by shading, sun illumination or variations in thin snow cover – we also must consider the effects of the RGB camera's automatic settings (ISO, aperture, shutter speed). These settings adjusted in response to changing scene features and illumination. Although we applied an averaging blending mode to reduce exposure inconsistencies, this does not fully eliminate differences in camera response across flight legs. Such factors may contribute to the spread in the red band values and affect correlation strength. However, in areas with high sediment concentrations, we expect their impact on red band "brightness" to be minimal. As in the previous section, we assess the correlation between surface temperature and red band values for the "entire field" dataset (Fig. 10b), for the "sediment" dataset (Fig. 10e) and for the "no sediment" dataset (Fig. 10h).





For the "entire field" data set, surface temperature and red band intensity show a moderate correlation ( $r_s = 0.35$ , Table A1). The corresponding scatterplot (Fig. 10b) reveals two clusters, which we disentangle by analyzing the "sediment" and "no-sediment" subsets separately. The "sediment" dataset shows a weak yet significant negative correlation between the red band intensity (lower values represent "darker" pixels) and surface temperature ( $r_s = -0.23$ , Fig. 10e). This pattern is consistent with expectations, as impurities typically increase radiation absorption (Ledley and Pfirman, 1997; Réveillet et al., 2022).

Surprisingly, the "no sediment" dataset showed the same moderate, but positive correlation ( $r_s = 0.4$ , Fig. 10h) between red band values and surface temperature as observed for the "entire field" dataset. In this case, darker areas were colder – opposite to the relationship found in the "sediment" dataset. This result was initially puzzling, since even under the consideration that clean snow most likely contains small amounts of impurities, there is no clear reason why lower red band values should correspond to lower temperatures.

A closer look suggests that variations in red band intensity within the "no sediment" dataset are primarily driven by differences in solar exposure and illumination, caused by small-scale topography such as snow dunes (Fig. 7d, e, f). In this case, the positive correlation reflects the influence of solar illumination and shading: shaded areas appear darker and colder, while sunlit areas are brighter and warmer. By contrast, in the "sediment" dataset, red band values capture both sediment concentration (which drives a negative correlation with surface temperature) and local irradiance variations (which drive a positive correlation). The overlap results of these competing influences yield the weak overall correlation ( $r_s = -0.23$ ). Nevertheless, the negative relationship dominates, indicating that in regions with higher sediment concentrations, the warming effect of impurities outweighs variations caused by local irradiance.

**Figure 9.** Scatter plot showing the relationship between slope and irradiance for the entire field, excluding stations and targets (a). (b) Histogram for the irradiance, the slope (c) and the aspect (d), each across the CS test field.

We can now interpret the pattern in Fig. 10b as the combined effect of the "sediment" and "no sediment" dataset. The positive correlation for the "entire field" dataset is largely driven by the "no sediment" (clean snow) areas, since the "sediment" dataset is about 20 times smaller and contributes little to the overall statistics. The stronger correlation in the "no sediment" dataset indicates that, in snowpacks without (visible) sediment deposition, surface temperature is controlled more directly by irradiance and shading than by the likely small "imperceptible" impurities. Thus, for clean snowpacks without significant sediment deposition, our results emphasize the dominant role of solar illumination and local topographic shading in driving surface temperature and, by extension, the near-surface energy balance.

In summary, we find that the correlation between red band values and surface temperature is stronger than the correlation between the snow depth proxy and surface temperature across all three sub-datasets. This indicates that the red band intensity – influenced by 1) impurity concentration, 2) illumination variations, and 3) optical snow depth differences – serves as a better predictor of local surface temperatures. As shown in the previous subsection, snow depth variations are not the primary driver. The contrasting correlations between the "no sediment" dataset ( $r_s = 0.4$ ) and the "sediment" dataset ( $r_s = 0.23$ ) suggest that local illumination conditions, driven by small-scale surface topography, play a major role in shaping surface temperatures where sediment is absent.

#### 3.2.3 The role of irradiance



Building on the previous section's findings that solar irradiance may play an important role in driving local temperature differences, we examined the correlation between modeled, topography-dependent irradiance and surface temperatures across all three sub-datasets (Fig. 10c, f, i). Each sub-dataset shows a significant, though weak, positive correlation between surface temperature and irradiance ("entire field": r<sub>s</sub> = 0.22, "sediment": r<sub>s</sub> = 0.22, "no-sediment": r<sub>s</sub> = 0.24). While the mean irradiance is nearly identical across all three sub-datasets (Fig. 8d, h, l), the variation (STD) is largest in the "sediment" dataset, likely reflecting higher snow depth and a rougher surface (Fig. 8e).

To investigate whether local irradiance influences the red band intensity (through illumination and shading) – which could explain the observed connection between red band values and surface temperature – we calculated the correlation between these two parameters. Across all three datasets, we found a moderate correlation between the red band values and irradiance (Table A1), confirming that variations in red band intensity are at least partially driven by irradiance (Fig. 11). In the "no sediment" dataset, the correlation between red band values and irradiance ( $r_s = 0.44$ ) is comparable to the correlation between red band values and surface temperature ( $r_s = 0.4$ ), supporting our hypothesis that irradiance is an important driver of surface temperature. While the correlation between red band values and irradiance is similar for the "sediment" dataset – because irradiance depends only on topography – sediment deposition has a stronger influence on surface temperature (Fig. 10e). This suggests that, in areas with sediment, impurities dominate surface heating, outweighing the effect of shading and local irradiance.

Figure 10. (a), (d) and (g) show the correlation between surface temperature and the snow depth proxy (the color bar is based on the red band values). (b), (e) and (h) show the surface temperature and irradiance (the color bar is based on the red band values). (c), (f) and (i) show the surface temperature and red band values (the color bar is based on the snow depth proxy). The first row is the "entire field" dataset, excluding targets and measurement stations. The second row is te "sediment" dataset, which only contains data points within areas of visible sediment deposition. The last row displays the "no sediment" dataset for areas of clean snow without visible sediment deposition. Dataset and correlation statistics details are Fig. A5 and in Table A1.

In Fig. 9a, we show the spread of modeled irradiance against the local slope at each pixel. Slope serves as a proxy for surface orientation, capturing variations in the local incidence angle and, consequently, in effective irradiance. The cluster of points around 71 Wm $^{-2}$  represents shaded areas receiving only diffuse shortwave radiation. The spatially averaged modeled irradiance – calculated using the measured snow topography (slope and aspect) to adjust the solar effective zenith angle – is  $592 \pm 45 \, \text{Wm}^{-2}$ , with a range from 71 to  $1167 \, \text{Wm}^{-2}$ . This modeled mean closely matches with the measured irradiance of  $593 \, \text{Wm}^{-2}$  at the time of calculation, as we used observed shortwave radiation to determine atmospheric transmissivity and the fraction of the diffuse radiation. Over the entire flight period (about 40 minutes), the temporal mean measured irradiance was  $593 \pm 10 \, \text{Wm}^{-2}$ .

Figure 11. Correlation between irradiance ( $Wm^{-2}$ ) and red band intensity for "entire field" dataset ( $r_s = 0.44$ ) (a), "sediment" dataset ( $r_s = 0.44$ ) (b) and "no sediment" dataset ( $r_s = 0.44$ ) (c). We cannot with certainty explain the horizontal cluster from about 140 to 160 red band intensity that stretches across a larger range of irradiances than the rest of the data. However, the only connection we found for this cluster is snow depth, because the cluster disappears for higher snow depths, as discussed in A5 (Fig. A6). This cluster, however, has a low data density (Fig. A6d) and is therefore outside of the scope of our study.

The uniform distribution of slope and aspects in the DEM (Fig. 9c, d) indicates a near-random terrain pattern, with no preferred alignment of roughness features. This explains the approximately normal distribution of the locally modeled irradiance (Fig. 9b). While the measured irradiance – representing irradiance on a flat plane – and the spatially average modeled irradiance (Fig. 9a, red and blue lines, respectively) are nearly identical, reflecting that the total incoming energy over the area remains constant, accounting for topography reveals a much wider range of values. The min-max spread and variability (STD) in modeled irradiance increase substantially when surface topography is considered, highlighting the strong influence of small-scale topography on local irradiance patterns.


Even at our relatively flat site, topography-dependent irradiance exhibits spatial variability (Fig. 9) with 95% of values ( $\pm 2$  STD) ranging from 502 to 682 Wm<sup>-2</sup> – roughly  $\pm 20\%$  of the areal mean. This variability arises from seemingly small changes in local topography, which primarily alter the apparent solar zenith angle and thus affect local surface temperatures (Wiscombe and Warren, 1980). While these small-scale topographic variations do not change the mean

areal irradiance – since the total incoming energy over the area remains constant – they can nonlinearly influence the local snowpack energy balance (Weller, 1969; Hao et al., 2022), producing localized temperature gradients both vertically and horizontally. For example, temperature differences between Sun-exposed and shaded sides of snow dunes can drive lateral heat fluxes (Sturm and Holmgren, 1994), modifying snow microstructure and affecting albedo and thermal conductivity (Clemens-Sewall et al., 2024). In more extreme cases, this lateral heat transfer and associated metamorphism can alter snow height and surface topography through sublimation (Gow, 1965; Orheim, 1968; Weller, 1969). Later in the season, as surface temperatures approach 0 °C, even small variations in local irradiance may trigger differential melting, freezing and densification, further amplifying spatial variability within the snowpack.

Moreover, surface topography – even from modest slopes – can distort the spectral shape of albedo via multiple reflections, leading to errors in estimating intercepted solar radiation (Picard et al., 2020; Manninen et al., 2021). Slopes as small as 1° can produce such distortions, with their complexity increasing with steeper terrain. Accurate albedo estimates, therefore, require accounting for surrounding topography at the full range of spatial scales (Picard et al., 2020; Manninen et al., 2021; Larue et al., 2020).

# 3.2.4 Integrating predictors via Linear Mixed-effects Modeling (LMM)





To complement the pairwise correlation analyses and better quantify the relative influence of snow depth, irradiance, and sediment, we combined the "sediment" and "no sediment" datasets (626,869 data points) and applied a linear mixed-effects model. This approach allows us to assess the independent contribution of each predictor to surface temperature while accounting for grouping effects in the data. The LMM provides a formal, integrated evaluation of the drivers identified in the previous sections and enables a comparison of their practical impact over the observed range of variability.

The LMM results (Table 3) confirm and quantify the patterns observed in the previous correlation analyses. All three predictors - snow depth, irradiance, and red band intensity – were significant, with irradiance producing the largest potential change in surface temperature ( $\Delta T \approx +4.3$  °C), followed by red band reflectance ( $\Delta T \approx -3.1$  °C) and snow depth ( $\Delta T \approx +0.75$  °C). The results remain consistent even when accounting for propagated errors in snow depth and surface temperature ( $\pm$  0.045 m and  $\pm$  2.41 °C respectively). These results reinforce our earlier findings: while snow depth has a measurable effect on surface temperature, the dominant controls on local surface temperature are irradiance and sediment (as captured by red band values), consistent with the spatial patterns and correlations discussed in the preceding sections. However, it needs to be kept in mind that irradiance and red band intensity are not independent parameters, because, obviously, shaded areas are darker than sun-exposed areas.

# 3.2.5 Technical considerations for high-resolution mapping of snow topography and snow surface temperature

Producing accurate maps of these variables requires three key technical considerations:

**Table 3.** LMM results for the prediction of snow surface temperature (predictors: snow depth proxy, irradiance, red band). The model included 626,869 observations across two datasets ("sediment" and "no sediment") with group variance = 0.292. Model fit: Restricted Maximum Likelihood (REML), variance of error term = 0.2916. All predictors were significant (p < 0.001).  $\Delta T$  shows the impact of each predictor over its observed range;  $\beta_{mean}$  and  $\Delta T_{noise}$  show results after introducing  $\pm$  0.045 m error in snow depth and  $\pm$  2.41 °C error in temperature (Monte Carlo, 100 iterations).

| Predictor        | Coeff ( $eta$ ) | $Coeff_{mean}$ ( $eta_{mean}$ ) | Range                  | <b>Δ</b> Τ (℃) | $\Delta T_{\text{noise}}$ (°C) |
|------------------|-----------------|---------------------------------|------------------------|----------------|--------------------------------|
| Snow depth proxy | 1.77            | 1.16                            | 0.42 m                 | 0.75           | 0.74                           |
| Irradiance       | 0.004           | 0.004                           | $1067\mathrm{Wm^{-2}}$ | 4.27           | 4.70                           |
| Red band         | -0.023          | -0.023                          | 134                    | -3.08          | -3.14                          |

- Precise georeferencing is essential for reliable spatial accuracy, requiring high-accuracy GPS and a sufficient number of visible and thermal targets to co-locate RGB and thermal images. We recommend at least 10 GCPs for a study site of 200 m x 200 m.
- 2. Rigorous post-processing of TIR images is critical to ensure accurate temperature data. We developed a correction algorithm that automatically detects and reduces temperature jumps caused by NUC correction events, minimizing sensor temperature drift and improving data accuracy. Our algorithm can be applied to various camera models, offering broad utility for studies beyond polar research.




- 3. Ground-based temperature measurements are necessary to validate airborne temperature surveys. Airborne temperature measurements should ideally be taken near local solar noon to capture peak solar radiation, minimize radiation variability, and maintain a high Sun angle to reduce shading. Clear skies are preferable to avoid cloud shadows that alter the surface temperature, though consistent high cloud cover can also be acceptable. The "best" conditions depend on the specific purpose of the survey, but random intermittent cloud cover should always be avoided as it is problematic.
  - At least two non-contact infrared radiometers (e.g., Apogee sensors or equivalent) should be set up during the flight to measure snow surface temperature and calibrate the airborne measurements. To ensure accurate calibration, there should be a significant temperature difference between targets (e.g. a hot target like sediment, sea ice or a temperature-controlled target with known emissivity, and a cold target like clean snow).
- 4. Ground-based snow depth measurements are necessary to validate and bias-correct airborne snow depth surveys (on flat sea ice) and calculate the snow depth proxy. These measurements should be taken in the flight area, though not necessarily during flight.

A balance between area coverage and flight duration was necessary to avoid changes in local irradiance (e.g., the measured irradiance during the entire flight increased from 575 to 609 Wm<sup>-2</sup>). Validation was challenging due to the non-uniform snow cover and the imprecision of the MagnaProbe measurement positiong. We used the MagnaProbe

measurements to correct the offset of the airborne DEM (Fig. 5), an approach that is valid primarily because of the small range of variability in the sea ice thickness and the smoothness of the ice surface. Our findings are based on a test site with smooth, approximately 2.4 m thick sea ice and a relatively flat snowcover of  $0.1 \pm 0.04$ m, representing a limited spatial scale. In areas with heterogeneous sea ice thickness and greater surface roughness – such as pack ice (Haas, 2004) or ridged ice (Itkin et al., 2023) – the DEM would reflect the combined topography of variable sea ice thickness, ridging, and snow depth, and our snow depth proxy would not be applicable. Future studies should extend these methods to more complex ice conditions and evaluate their sensitivity in energy balance models. High-resolution UAV observations remain a valuable tool for improving our understanding of snow-ice interactions and informing both regional models and satellite-based remote sensing applications.

# 4 Summary and Conclusions







This study demonstrates that UAV-based remote sensing provides an effective approach for mapping snow topography, depth and surface temperatures on relatively uniform landfast sea ice in the Ross Sea, Antarctica. A key technical advance is our algorithm to correct NUC camera calibration events, which ensures accurate airborne temperature measurements and has broader applicability beyond polar research.

Our high-resolution data reveal how small-scale spatial variability in snow surface topography strongly influences irradiance and surface temperatures. Assuming a flat ice surface underestimates irradiance variability, potentially affecting non-linear energy balance processes such as localized melting and sediment transport. We found that irradiance and surface impurities dominate surface temperature patterns, while snow depth plays a secondary role.

Although our snow depth proxy works only on flat, smooth sea ice, the methods for retrieving topography and surface temperature are valid for a wider range of ice types. Applying these techniques to heterogeneous or deformed sea ice will require careful adaptation but could provide critical insights into local energy balance dynamics. Overall, our work highlights the feasibility and importance of high-resolution airborne measurements for understanding snow on sea ice processes and the potential to improve snow parameterizations in climate models and remote sensing retrievals.

We demonstrate that if models do not account for snow surface roughness – either due to the lack of availability of local topography data or insufficient model resolution – they tend to underestimate the variability in irradiance and simplify the amount of energy available for localized snow metamorphism and melt processes, particularly in areas with pronounced topography variations.

Taken together, our findings highlight that both small-scale topography and surface impurities are key controls on local snow surface energy balance, and their combined effects should be carefully represented in models to improve predictions of snowpack evolution and sea ice melt.

# Appendix A: Additional figures and statistics

**Figure A1.** Tidal movements (m) at the Scott Base tidal gauge from the sea level recorder (nitrogen bubbler system with a paroscientific pressure transducer located on a bottom-mounted spigot of the osmosis boom and barometer). The data is recorded in 5 minute intervals. The orange part of the plot marks the flight survey.

Figure A2. (a) Hourly averaged air temperature and wind speed during field campaign at CS test field. The vertical yellow bar indicates the UAV mission. (b) The prevailing wind directions at the CS test field during the field campaign.

**Figure A3.** Sea ice surface at two different snowpit locations (AS006, AS002) inside the 200 m x 200 m CS test field during our field campaign showing the flat and smooth sea ice surface.

**Figure A4.** Distribution of offset-uncorrected (black) and offset-corrected (gray) elevation within a 2.5 m radius of six randomly selected MagnaProbe locations. The red dashed line indicates the snow depth measured with the MagnaProbe, and **n** is the number of pixel values from the DEM within the 2.5 m radius. The variability in the DEM-derived elevation/snow depth proxy within the radius around the MagnaProbe is almost as much as in the whole field.

Figure A5. Masks used to create the three sub-data sets "entire field" (a), "sediment" (b) and "no sediment" (c) in the RGB orthomosaic.

**Figure A6.** (a) scatter plot of red band intensity versus snow depth proxy (m) for "no sediment" sub-dataset ( $r_s = 0.08$ ). The red line indicates the visible cluster separation. (b) Scatter plot of red band intensity versus irradiance ( $Wm^{-2}$ ) for "no sediment" dataset, considering only data points where the snow depth proxy exceeds 0.25 m ( $r_s = 0.69$ ). (c) Same as (b) but for snow depth proxy values below 0.25 m ( $r_s = 0.41$ ). Statistics details are provided in TableA1. These plots reveal that the horizontal cluster observed the red band intensity range of 140 to 160 disappears when considering only snow depth proxy values is connected to the snow depth as it disappears when considering only snow depth proxy values above 0.25 m, indicating a connection to snow depth. For snow depth proxy values below 0.25 m, the entire irradiance range is present, suggesting a link to topography. However, the red band intensity shows no direct dependence on the snow depth proxy, and a more detailed analysis is beyond the scope of this study. (d) Highlights that the horizontal extent of the cluster corresponds to a region with low data density.

**Table A1.** Statistics table for Fig. 10. Entire field: all data points excluding targets and measurement stations. Sediment: data points within areas of visible sediment deposition (and clean snow), excluding targets and measurement stations in these areas. No sediment: areas with no visible sediment deposition, excluding targets and measurement stations. The three datasets are displayed in Fig. A5. The correlation strengths are formatted as plain (very weak), *italic* (weak), **bold** (moderate), and underlined (strong) font styles.

| Parameter A | Parameter B                 | Data points | Spearman (r <sub>s</sub> ) | p-value ( $\alpha$ = 0.01) | Info         |
|-------------|-----------------------------|-------------|----------------------------|----------------------------|--------------|
| Temperature | Snow depth proxy            | 6211943     | 0.16                       | <1e-16                     | entire field |
| Temperature | Snow depth proxy            | 285396      | 0.26                       | <1e-16                     | sediment     |
| Temperature | Snow depth proxy            | 341473      | 0.06                       | <1e-16                     | no sediment  |
| Temperature | Red Band                    | 6211943     | 0.35                       | <1e-16                     | entire field |
| Temperature | Red Band                    | 285396      | - 0.23                     | <1e-16                     | sediment     |
| Temperature | Red Band                    | 341473      | 0.40                       | <1e-16                     | no sediment  |
| Temperature | Irradiance                  | 6211943     | 0.22                       | <1e-16                     | entire field |
| Temperature | Irradiance                  | 285396      | 0.22                       | <1e-16                     | sediment     |
| Temperature | Irradiance                  | 341473      | 0.24                       | <1e-16                     | no sediment  |
| Red band    | Irradiance                  | 6211943     | 0.44                       | <1e-16                     | entire field |
| Red band    | Irradiance                  | 285396      | 0.41                       | <1e-16                     | sediment     |
| Red band    | Irradiance                  | 341473      | 0.44                       | <1e-16                     | no sediment  |
| Red band    | Snow depth proxy            | 341473      | 0.08                       | <1e-16                     | no sediment  |
| Red band    | Snow depth proxy (> 0.25 m) | 16965       | 0.69                       | 
  - Webster, M., Gerland, S., Holland, M., Hunke, E., Kwok, R., Lecomte, O., Massom, R., Perovich, D., and Sturm, M.: Snow in the changing sea-ice systems, Nature Climate Change, 8, 946–953, https://doi.org/10.1038/s41558-018-0286-7, 2018.
- Weller, G.: The Heat and Mass Balance of Snow Dunes on the Central Antarctic Plateau, Journal of Glaciology, 8, 277-284, 795 https://doi.org/10.3189/s0022143000031257, 1969.
  - Wendisch, M., Brückner, M., Crewell, S., Ehrlich, A., Notholt, J., Lüpkes, C., Macke, A., Burrows, J. P., Rinke, A., Quaas, J., Maturilli, M., Schemann, V., Shupe, M. D., Akansu, E. F., Barrientos-Velasco, C., Bärfuss, K., Blechschmidt, A.-M., Block, K., Bougoudis, I., Bozem, H., Böckmann, C., Bracher, A., Bresson, H., Bretschneider, L., Buschmann, M., Chechin, D. G., Chylik, J., Dahlke, S., Deneke, H., Dethloff, K., Donth, T., Dorn, W., Dupuy, R., Ebell, K., Egerer, U., Engelmann, R., Eppers, O., Gerdes, R., Gierens, R., Gorodetskaya, I. V., Gottschalk, M., Griesche, H., Gryanik, V. M., Handorf, D., Harm-Altstädter, B., Hartmann, J., Hartmann,
- 800

- M., Heinold, B., Herber, A., Herrmann, H., Heygster, G., Höschel, I., Hofmann, Z., Hölemann, J., Hünerbein, A., Jafariserajehlou, S., Jäkel, E., Jacobi, C., Janout, M., Jansen, F., Jourdan, O., Jurányi, Z., Kalesse-Los, H., Kanzow, T., Käthner, R., Kliesch, L. L., Klingebiel, M., Knudsen, E. M., Kovács, T., Körtke, W., Krampe, D., Kretzschmar, J., Kreyling, D., Kulla, B., Kunkel, D., Lampert, A., Lauer, M., Lelli, L., von Lerber, A., Linke, O., Löhnert, U., Lonardi, M., Losa, S. N., Losch, M., Maahn, M., Mech, M., Mei, L., Mertes,
  S., Metzner, E., Mewes, D., Michaelis, J., Mioche, G., Moser, M., Nakoudi, K., Neggers, R., Neuber, R., Nomokonova, T., Oelker, J., Papakonstantinou-Presvelou, I., Pätzold, F., Pefanis, V., Pohl, C., van Pinxteren, M., Radovan, A., Rhein, M., Rex, M., Richter, A., Risse, N., Ritter, C., Rostosky, P., Rozanov, V. V., Donoso, E. R., Saavedra Garfias, P., Salzmann, M., Schacht, J., Schäfer, M., Schneider, J., Schnierstein, N., Seifert, P., Seo, S., Siebert, H., Soppa, M. A., Spreen, G., Stachlewska, I. S., Stapf, J., Stratmann, F., Tegen, I., Viceto, C., Voigt, C., Vountas, M., Walbröl, A., Walter, M., Wehner, B., Wex, H., Willmes, S., Zanatta, M., and Zeppenfeld, S.:
  Atmospheric and Surface Processes, and Feedback Mechanisms Determining Arctic Amplification: A Review of First Results and Prospects of the (AC)3 Project, Bulletin of the American Meteorological Society, 104, https://doi.org/10.1175/bams-d-21-0218.1, 2022.
  - Wever, N., Leonard, K., Maksym, T., White, S., Proksch, M., and Lenaerts, J. T.: Spatially distributed simulations of the effect of snow on mass balance and flooding of Antarctic sea ice. Journal of Glaciology, 67, 1055–1073, https://doi.org/10.1017/jog.2021.54, 2021.
- Wigmore, O. and Molotch, N. P.: Weekly high-resolution multi-spectral and thermal uncrewed-aerial-system mapping of an alpine catchment during summer snowmelt, Niwot Ridge, Colorado, Earth System Science Data, 15, 1733–1747, https://doi.org/10.5194/essd-15-1733-2023, 2023.
  - Wigmore, O., Mark, B., McKenzie, J., Baraer, M., and Lautz, L.: Sub-metre mapping of surface soil moisture in proglacial valleys of the tropical Andes using a multispectral unmanned aerial vehicle, Remote Sensing of Environment, 222, 104–118, https://doi.org/10.1016/j.rse.2018.12.024, 2019.

- Wiscombe, W. J. and Warren, S. G.: A model for the spectral albedo of snow. I: pure snow., Journal of the Atmospheric Sciences, 37, 2712–2733, https://doi.org/10.1175/1520-0469(1980)037<2712:AMFTSA>2.0.CO;2, 1980.
- Zatko, M. C. and Warren, S. C.: East Antarctic sea ice in spring: Spectral albedo of snow, nilas, frost flowers and slush, and light-absorbing impurities in snow, Annals of Glaciology, 56, 53 64, https://doi.org/10.3189/2015AoG69A574, 2015.
- Zhou, X., Li, S., and Morris, K.: Measurement of all-wave and spectral albedos of snow-covered summer sea ice in the Ross Sea, Antarctica, Annals of Glaciology, 33, 267–274, https://doi.org/10.3189/172756401781818743, 2001.