# Peer review of "How flat is flat? Investigating snow topography and the spatial variability of snow surface temperature on landfast sea ice using UAVs in McMurdo Sound, Antarctica"

_EGUsphere, 2025_

## Referee Comment (RC2)

**Review of " How flat is flat? Investigating the spatial variability of snow surface temperature and roughness on landfast sea ice using UAVs in McMurdo Sound, Antarctica"**

**General Comments**

This study presents an innovative UAV-based approach to mapping snow surface temperature and topography on Antarctic landfast sea ice. The authors provide detailed documentation of their UAV flights, ground-based measurements, and image processing steps, including a novel algorithm to correct thermal drift due to NUC (Non-Uniformity Correction) events in the thermal camera.

The study is technically sound and methodologically robust, offering high-resolution spatial insights into processes that are often oversimplified in energy balance models. The authors convincingly demonstrate that sediment and irradiance variability, rather than snow depth, dominate surface temperature patterns in their study area.

However, the manuscript in its current form is excessively long and includes some minor technical and formatting inconsistencies. Furthermore, the broader applicability of the results are not enough. With revision, this manuscript has strong potential for publication and contributes valuable observational data and methodology to polar remote sensing and surface energy balance research.

**Major comments**

1. My first major comments is the primary hypothesis, that snow depth drives spatial variability in snow surface temperature, is tested but ultimately unsupported. For example, the correlation between snow depth and surface temperature is weak (e.g., Spearman's rs = 0.16 for the entire field), which the authors acknowledge. While the discussion then shifts toward the influence of sediment and irradiance, the analysis remains largely qualitative. A more rigorous statistical treatment, such as a multivariate regression analysis that accounts for snow depth, sediment presence (e.g., red-band reflectance), irradiance, and roughness, would be necessary to better disentangle the relative contributions of each factor. Moreover, the identification of sediment patches is based solely on manual inspection of RGB images and red band intensity. This approach is subjective and sensitive to camera auto-settings and changing illumination conditions. As a result, I have serious concerns about the accuracy and reproducibility of this classification. Could the authors develop an objective spectral threshold or a sediment index to quantify contamination levels? Could radiative transfer modeling or image normalization techniques help constrain uncertainty? How transferable is this method to other optical datasets or regions? As it stands, the interpretation of albedo-related heating effects remains qualitative and lacks the quantitative rigor needed to support strong conclusions. This weakens the scientific impact of the paper.

2. While it is perfectly valid to report a null result (i.e., snow depth is not the main driver of surface temperature variability), the manuscript does not provide a compelling quantitative alternative. Sediment and irradiance effects are proposed as alternative drivers, but again, these are explored only descriptively, without any systematic modeling framework. There is no attempt to define a sediment contamination index or

perform a statistical regression that incorporates the full range of possible explanatory variables. As a result, the conclusions remain speculative and unsupported by robust evidence. This lack of quantitative follow-up significantly limits the paper's strength and broader relevance.

3. My second major concern is the very limited spatial and temporal scope of the study, which raises questions about the broader significance and generalizability of the results. The entire analysis is based on a single 200 × 200 m region of landfast sea ice, under relatively flat and low-wind conditions. This kind of super-local case study might be sufficient for a technical demonstration, but how representative is it of more heterogeneous, dynamic and any other Antarctic sea ice environments? Can any of the findings, especially the role of sediment or microtopography in surface temperature variability, be extended to larger spatial scales or different surface types? What about temporal representativeness? The study captures only one moment in time under clear-sky conditions. Without any diurnal or seasonal coverage, I found the conclusions are difficult to generalize. Have the authors considered applying the same methods to a second site, or repeating flights under different light/wind conditions? If not, this should be clearly acknowledged as a limitation.

4. Additionally, the snow depth proxy derived from the UAV DEM rests on the assumption of a flat ice surface and uses a single calibration offset from ground-based measurements. But where is the validation? There is no high-precision GNSS positioning of the MagnaProbe data, and the ±0.05 m uncertainty in snow depth is not propagated into any of the correlation analyses or surface temperature interpretations. How does this uncertainty affect the reliability of the snow temperature correlation results? Are there areas where DEM uncertainty dominates the signal? This aspect of the study is underdeveloped and undermines confidence in one of its core data layers.

5. A further issue is the lack of an overall uncertainty framework. The authors had ±2.1°C for temperature correction (Figure 4), but there is little analysis of how this uncertainty varies spatially (e.g., near image edges, in shaded areas), or how it might interact with surface roughness, sediment detection, or irradiance calculation. What are the sources of uncertainty from lens effects, and how do they affect the final interpretation? Could they lead to systematic biases in warmer sediment zones or sloped surfaces?

6. There is also a surprising absence of any actual surface energy budget analysis. The paper frequently references "energy balance" but never attempts to break it down into its components (e.g., incoming solar, reflected solar, outgoing longwave, conduction). Without this, the proposed mechanisms, e.g., sediment warming or irradiance enhancement on slopes, remain less clearly. Why not use the DEM and albedo proxies to estimate slope-corrected irradiance per pixel, or combine reflectance and temperature into a basic radiative balance estimate?

7. Finally, the overall discussion remains descriptive and lacks synthesis. What is the key physical message here? Are the findings consistent with prior studies in the Antarctic? What is new about the relationship between microtopography, sediment, and surface temperature, and what does it imply for satellite-based retrievals or sea ice models? Without clearer integration of results into a broader scientific context, the paper risks reading more like a high-resolution mapping exercise than a process-oriented scientific study.

**Specific comments:**

1. Section 2.3.3: The snow depth proxy construction is hard to follow. Are you aligning in situ snow depth data with the DEM by GPS coordinates, or using a

stereophotogrammetric offset? Please clarify the procedure with a schematic or simple diagram. This section needs better structure to explain how the reference elevation is selected and how uncertainty is estimated.

2. Figure 8: (1) The latitude/longitude labels are unconventional and difficult to interpret; (2) The color bar is unintuitive (blue implies deep snow, which is opposite of expectations). Consider reversing the color scale or annotating it more clearly. (3) Does the 0 m value mean bare ice? Please explain this in the caption.

3. Figures 10 and 13. These are not well-integrated into the discussion and don't provide substantial new insight. Consider removing them or clearly stating their purpose.

4. Line 300:"...to/by 0.53 °C?". Please clarify whether this value is a change "by" or an absolute value "of".

5. Line 486: "...area of 200 m.". This is not an area. You probably mean "length of 200 m" or "area of 200 m × 200 m".

6. Line 332 and Line 568: remove one "the"

7. Figure A2 and A3 should be swapped.

8. Figure A4 (b) and (c) add little value, it's redundant with (a). Consider removing or merging.

---

## Author Comment (AC1)

**01 Review**

*General*

The authors investigate how snow distribution patterns affect the surface temperature of snow on Antarctic sea ice, a key but under-studied factor in the sea-ice energy balance. The introduction puts the topic of the paper well into context and provides a great overview of the current state-of-the-art. To contribute to this vast topic of polar sea ice, the authors applied UAV and ground measurements to create high-resolution maps of snow topography and surface temperature over uniform landfast sea ice in McMurdo Sound, Antarctica. The measurement site, the ground-based and airborne methods are introduced in detail. Regarding the latter, a novel algorithm was developed to correct thermal drift in UAV thermal imagery, ensuring consistent temperature data. Based on these maps and correlations, the authors investigated the reasons for observed surface temperature variations. As a result, the surface temperature anomalies were mainly linked to visible sediment on the snow, not snow depth, which has been the authors initial hypothesis. They further found that small-scale topography significantly affected local solar irradiance, and assuming uniform irradiance underestimated its variability. Overall, sediment and irradiance were found to have a stronger influence on snow surface temperature than snow depth, highlighting the importance of surface features in energy balance modelling on sea ice.

The manuscript provides an important contribution to the analysis of polar surface temperature variations. It presents interesting and valuable results, which help to identify gaps in common surface temperature retrievals assuming flat surfaces. It highlights the problems and mismatches they struggle with and introduces proper solutions. I highly recommend its publication after the authors have revised the manuscript regarding the comments listed below.

*Major comment*

**Length:** The paper is very long, which makes it difficult to keep the readers attention from the beginning to the end. However, I think there is some potential to shorten the paper significantly.

1. In my opinion, there is no reason to separate the results and discussion sections. On the contrary, the separation results in a lot of repetition, which unnecessarily lengthens the paper. By merging the two sections, this could be avoided, and the paper could be significantly shortened, which would also make it more focused. The individual sections already have similar topics, so this should be easy to do.

AC: We thank the reviewer for this valuable suggestion and agree that merging the Results and Discussion sections will help reduce redundancy, improve clarity, and enhance the overall focus of the manuscript. We will follow this suggestion and restructure these sections accordingly in the revised version.

2. There are several graphs that represent more or less the same thing. For example, Fig. 1a,b and Fig. 5 a,b. One figure, either a or b, would be sufficient here. Some figures can be merged. For example Figure 8 and 9. Why not have one column with the Red Band Value (G and B are not really used), one column for the DEM and one for the temperature? Furthermore, there are graphs that in my opinion are not needed at all. For example, Figures 10 and 13. I have also made additional suggestions under Minor and Technical Corrections.

AC: We agree with the reviewer's assessment and are going to revise the figure layout accordingly. Specifically, we are going to adjust Fig. 1, streamline Fig. 5 by merging it with Fig. 4 as a third panel. We are also going to combine Figs. 8 and 9 into a single figure with three columns (Red band, DEM, TIR) to enhance readability. Lastly, we are going to move Figs. 10 and 13 to the appendix, as they provide useful background but are not essential to the main narrative.

3. Some further sections can be skipped as own sections and merged into others. For example, the main message from 2.3.4 fits to the introduction of the camera, Section 4.5 belongs to the summary and conclusion part.

AC: We agree with the reviewer and are going to merge Section 2.3.4 into Section 2.3.5 to improve structure and flow. We are also going to move Section 4.5 into the Summary and Conclusions section, as it aligns better thematically.

**Figures:** The figures are often not really introduced, but are only mentioned in brackets after certain statements, so that the reader has to find out for himself what is shown. This makes it difficult to read fluently and understand directly. It would be good to describe what is shown in the text with one or two sentences.

AC: We agree with the reviewer and are going to revise the manuscript to more clearly introduce and describe each figure within the main text. This will improve readability and help the reader better understand the content and relevance of each figure.

***Minor comments:***
1. Order: Figures are partly not numbered in the order they are used in the text.

AC: We acknowledge this oversight and are going to revise the manuscript to ensure that all figures are numbered in the order in which they are referenced in the text.

2. Space signs: Space signs are multiple times missing between numbers and units, between figure abbreviation and figure number and in front of citations that are given in brackets.

AC: We acknowledge this issue and are going to correct all spacing inconsistencies throughout the manuscript, including those between numbers and units, figure abbreviations and figure numbers, as well as spacing before in-text citations.

3. Indices and units: Indices are sometimes written in italic letters and sometimes in non-italic letters. For reasons of consistency, you should write all indices in non-italic letters.

AC: We are going to standardize the formatting of all indices in the manuscript, ensuring they are written in non-italic letters for consistency.

4. P2, L46: What means the original hypothesis was? It should stay the same, although it might have been rejected.

AC: We understand the reviewer's concern regarding the phrasing. We will clarify the language to indicate that the original hypothesis remains as stated but that our results do not support it. This will ensure the hypothesis is clearly presented and its evaluation is transparent.

5. Sect. 2.3.2: You might have to revise this section to make it more clear to the reader what happens here. I had to read it several times and I am still not sure if I got it right. Do you match the in situ snow depth measurements by their GPS position into the RGB images? Or is it some kind of a stereographic method? Please revise it to make this easier understandable. Maybe also a sketch might help to better understand the procedure behind.

AC: We appreciate the reviewer's effort to understand this section. We believe the comment may refer more to Section 2.3.3, where we use the DEM rather than RGB images to relate in situ snow depth measurements. Nonetheless, we recognize the current explanation could be clearer and potentially confusing. We will revise this section to provide a more straightforward description of the methodology, including the spatial matching approach between the DEM and in situ measurements, and consider adding a schematic to aid reader comprehension.

6. Sect. 2.3.4: I don't think that this section is really needed. It should be enough, if you mention the conversion within one sentence, when you introduce the camera.

AC: We are going to merge Section 2.3.4 with Section 2.3.5, as suggested above, and will condense the content to a brief mention of the conversion when introducing the camera.

7. Sect. 2.3.5: This is a very well-thought-out method, which leads to convincing results. I just wonder how you derive the absolute calibration. Is it a predefined function for each pixel from the lab, which is then scaled for each pixel by the NUC? Or do you know the temperature of the shutter (which then acts like a black body) and in parallel is used as a homogeneous target to remove the non-uniformity of the single pixels? Furthermore, later in Line 305 you discuss a vignette effect, which originates from the lens properties. That brings me to the question where the shutter is installed. Is it installed in front of the lens or between the lens and the detector? For the latter you will imprint the structure of the lens-own temperature into the images, when you perform the NUC, which then leads to this vignette effect and might also changes the absolute calibration.

AC: Thank you for the insightful questions. Unfortunately, the camera's internal calibration process is not fully accessible to us, and detailed information on the absolute calibration or shutter placement is not provided by the manufacturer (investigation with DJI was unfortunately unfruitful). The vignette effect is taken care by applying an oval mask as part of the AGISOFT Metashape processing workflow (see Fig. 3 ).

8. Sect. 3: The initial paragraph is a repetition and can be skipped.

AC: We agree with the reviewer and will remove the initial paragraph from Section 3 to avoid repetition.

9. P24, Fig 12: It would be great to have such a "correlation" plot for "degree of sedimentation" vs. temperature. I guess here you would find a significant high correlation. Of course, I see the point with the difficulties related to the varying camera settings as you write in Line 337 and the shadows you mention in Line 408. But shouldn't be the first issue solved by your NUC calibration? Furthermore, you could use your DEM to extract regiones, which might be affected by shadows. Adapting both, you would be able to select scenes, which are unaffected by shadows and after a normalization you can extract an arbitrary value for the degree of sedimentation, which might range from 0 to 1.

AC: We appreciate the suggestion, and the level of detail that the reviewer has thought about this; however, the 'degree of sedimentation' is derived from RGB imagery, which is not corrected by the thermal NUC calibration. Since RGB values are affected by varying illumination and camera auto-settings, they do not provide consistent surface reflectance measurements, and even if we remove the totally shaded areas, the issue of the

illumination remains. Therefore, developing a rigorous, normalized sedimentation index from these data is currently not feasible within the scope of this study.

10. Sect. 4.5: Belongs to summary and conclusions. There, it fits well at the very end.
AC: We agree with the reviewer and will move Section 4.5 to the Summary and Conclusions section, placing it at the end for better flow.

11. Sect. 5: I would expect some words regarding the correlations and the main findings resulting from them.
AC: We agree and will add a focused discussion on the correlations and highlight the main findings in Section 5.

**Technical comments**

*AC:* We thank the reviewer for the detailed technical and editorial suggestions. We are going to carefully review the manuscript to address all points raised. A few comments on points 17, 19, and 28:

- Point 17: We appreciate the reviewer's attention to detail. We have carefully considered this and note that temperature differences are consistently reported in degrees Celsius throughout the manuscript for clarity and consistency.
- Point 19: Thank you for pointing this out we are going to clarify that the RMSE for surface temperature decreased by 0.53 °C, down to 0.58 °C".
- Point 28 (Fig. 10): We believe this figure is valuable as it shows the range and distribution of calculated insolation values across the study area. While aspect itself may not be a primary focus, it provides context about solar exposure variability. We will clarify this in the text to improve its relevance and reader understanding.

1. Sun: "Sun" is a proper name and should be written with capital letter. It appears several times throughout the manuscript.
2. **P1, L13:** ... our study (200x200 m) area → ...our study area (200 x 200 m$^2$)
3. **P2, L26:** Wendisch et al. 2023 might be cited as well as it summarizes a huge project with its focus on Arctic amplification.
4. **P2, L53:** → ...(), colder
5. **P3, L78:** Acronym
6. **P3, L85:** Which infrared? Near-, thermal-, far-?

7. **P4, Fig. 1:** Only Fig. 1a is needed. Figure 1b does not offer to many more details. If you prefer to keep both, add E(ast) and S(outh) in Fig. 1b and draw (a) and (b) into the Figures. I have overseen it for a long time. Furthermore, there is a missing space sign in the figure caption before (yellow rectangle).
8. **P4, L92:** Add an outline at the end of the introduction?
9. **P5, L115:** Size and time period are already mentioned in the sentence before. Skip the repetition.
10. **P5, L129:** Italic index, which might be a typo. Also, two lines later.
11. **P6, Table 1, caption:** Skip "used in this paper". Should be clear.
12. **P6, Table 1:** Maybe include some empty lines in between the different parameters. Since some parameters use two lines it is hard to see, which entries belong to the same parameter.
13. **P6, L136:** This last sentence belongs two the first sentence of this section. You should move it there. It will fit better.
14. **P8, L190:** back → black
15. **P9, Fig. 2:** The transparency is almost not visible on a printed version.
16. **P9, L214:** Here you jump over six figures to show something in a figure, which is not yet introduced. Please avoid this.
17. **P10, L247:** Differences in Kelvin, not in °C
18. **P14, Fig. 5:** ... and boxplots (b) → ...and (b) boxplots; actually I think that only one of the figures is needed. They more or less show the same.
19. **P14, L300:** to or by 0.53°C?
20. **P15, L320:** Fig.6a → Fig. 6a and the same for Fig. 6b a few lines later
21. **P15, L326:** ... higher-resolution (10 s) sea... → ... temporally higher resolved (10 s) sea...
22. **P15, L338:** better to use radiation instead of light.
23. **P18, Fig. 8:** Here, I have several things. (i) Uncommon Lat/Long values. I don't know how to interpret them. (ii) The colour bar is unintuitive. To me blue colours give the impression of lower snow depth, but it is the other way round. (iii) The scale ranges from 0 m to 0.6 m. Does it mean that there are parts with bare ice in the images?
24. **P19, L 398:** Missing brackets for the citation.
25. **P19, 403:** skip "and a small fraction of clean snow." It will be said in Line 405.
26. **P19, L405:** QGIS?
27. **P19, L 408:** RGB and in the next line R/G/B
28. **P21, Fig. 10:** Not really introduced and not really needed. What is the aspect?
29. **P21, L432:** flight legs instead of flight lines?
30. **P23, L453:** can't → cannot
31. **P23, L469:** Don't start sentence with an abbreviation.

32. **P24, Fig. 12:** Would be helpful to add a name list to the single figures. Maybe as a fourth column left or right of the graph, were you just write (vertically) All, Sediment, and No-Sediment.
33. **P25, L486:** ... area of 200 m. → is no area
34. **P26, L500:** e.g. → e.g.,
35. **P27, L557:** ... traps the light is a not so well expression. Try to avoid it.
36. **P28, Fig. 14:** Add All, Sediment, No-Sediment to the figures.
37. **P28, L568:** ... the the ...
38. **P28, L575:** Missing space sign in front of citation. Appears three times more within this and the next paragraph.
39. **P30, L627:** viable → sustainable?
40. **P32, Fig. A3:** x axis of graph b is incomplete.
41. **P33, Fig. A4:** (a) and (b) are missing inside the graphs. However, graph (b) is not really needed. It is already well visible in (a).

**Reference:**

Wendisch, M., et al.: Atmospheric and Surface Processes, and Feedback Mechanisms Determining Arctic Amplification: A Review of First Results and Prospects of the (AC)3 Project, Bull. Amer. Meteorol., 104 (1), E208–E242, doi:10.1175/BAMS-D-21-0218.1, 2023.

---

## Author Comment (AC2)

**02 Review**

General Comments

This study presents an innovative UAV-based approach to mapping snow surface temperature and topography on Antarctic landfast sea ice. The authors provide detailed documentation of their UAV flights, ground-based measurements, and image processing steps, including a novel algorithm to correct thermal drift due to NUC (Non-Uniformity Correction) events in the thermal camera.

The study is technically sound and methodologically robust, offering high-resolution spatial insights into processes that are often oversimplified in energy balance models. The authors convincingly demonstrate that sediment and irradiance variability, rather than snow depth, dominate surface temperature patterns in their study area.

However, the manuscript in its current form is excessively long and includes some minor technical and formatting inconsistencies. Furthermore, the broader applicability of the results are not enough. With revision, this manuscript has strong potential for publication and contributes valuable observational data and methodology to polar remote sensing and surface energy balance research.

Major comments

1. My first major comments is the primary hypothesis, that snow depth drives spatial variability in snow surface temperature, is tested but ultimately unsupported. For example, the correlation between snow depth and surface temperature is weak (e.g., Spearman's rs = 0.16 for the entire field), which the authors acknowledge. While the discussion then shifts toward the influence of sediment and irradiance, the analysis remains largely qualitative. A more rigorous statistical treatment, such as a multivariate regression analysis that accounts for snow depth, sediment presence (e.g., red-band reflectance), irradiance, and roughness, would be necessary to better disentangle the relative contributions of each factor. Moreover, the identification of sediment patches is based solely on manual inspection of RGB images and red band intensity. This approach is subjective and sensitive to camera auto-settings and changing illumination conditions. As a result, I have serious concerns about the accuracy and reproducibility of this classification. Could the authors develop an objective spectral threshold or a sediment index to quantify contamination levels?

Could radiative transfer modeling or image normalization techniques help constrain uncertainty? How transferable is this method to other optical datasets or regions? As it stands, the interpretation of albedo-related heating effects remains qualitative and lacks the quantitative rigor needed to support strong conclusions. This weakens the scientific impact of the paper.

AC: We thank the reviewer for these thoughtful comments. We will attempt to include a multiple linear regression analysis using snow depth, irradiance, and red band values as

predictors of snow surface temperature. This will be carried out for both the targeted clean and sediment datasets, as well as the full test field. While we agree on exploring this path to provide an additional quantification of our results, we are going to explore the feasibility of this approach as the parameters are mostly dependent on each other and cross-correlations might be the limiting factor.

In the targeted dataset, we expect red band values to reflect sediment presence, which would overwhelm the illumination effects if sediment concentration were visible, such as is the case here. In the full field, however, the red band values are likely to capture broader effects such as illumination and shadowing rather than sediment alone. While we could remove the totally shaded regions through the irradiance modelling, the irradiance modelling will not account for the differences in illuminations in our images. Given the limitations of the available imagery, we do not consider the development of a more objective sediment index or the application of image normalization techniques feasible within the scope of this study.

We acknowledge the subjectivity involved in manually delineating sediment patches. However, we are not intending to quantify sediment concentrations, but merely to try and exclude areas with significant and visible sediments, so we can understand what effects temperatures of relatively clean snow. This study is intended as a proof of concept at a single site. Broader evaluation across different sites is planned for future work but lies outside the scope of the current paper.

2. While it is perfectly valid to report a null result (i.e., snow depth is not the main driver of surface temperature variability), the manuscript does not provide a compelling quantitative alternative. Sediment and irradiance effects are proposed as alternative drivers, but again, these are explored only descriptively, without any systematic modeling framework. There is no attempt to define a sediment contamination index or 2 perform a statistical regression that incorporates the full range of possible explanatory variables. As a result, the conclusions remain speculative and unsupported by robust evidence. This lack of quantitative follow-up significantly limits the paper's strength and broader relevance.

AC: We respectfully disagree with the characterization that our analysis lacks a quantitative framework. While some relationships are indeed weak, this reflects the nature of the data rather than a lack of quantitative effort. We disagree that a weak correlation is not a result, because it shows that, for example snow depth, is clearly not the dominant driver of surface snow temperatures. As with our response to the previous reviewer, we will address this point by including a multiple linear regression analysis incorporating snow depth, irradiance, and red band values as predictors of snow surface temperature. This approach will allow us to quantify the relative contributions of these variables beyond the current descriptive treatment. As previously noted, the development of a sediment

contamination index lies outside the scope of this study. The revised manuscript will include additional analysis to support and clarify our conclusions.

3. My second major concern is the very limited spatial and temporal scope of the study, which raises questions about the broader significance and generalizability of the results. The entire analysis is based on a single 200 × 200 m region of landfast sea ice, under relatively flat and low-wind conditions. This kind of super-local case study might be sufficient for a technical demonstration, but how representative is it of more heterogeneous, dynamic and any other Antarctic sea ice environments? Can any of the findings, especially the role of sediment or microtopography in surface temperature variability, be extended to larger spatial scales or different surface types? What about temporal representativeness? The study captures only one moment in time under clear-sky conditions. Without any diurnal or seasonal coverage, I found the conclusions are difficult to generalize. Have the authors considered applying the same methods to a second site, or repeating flights under different light/wind conditions? If not, this should be clearly acknowledged as a limitation.

AC: We acknowledge the limited spatial and temporal scope of this study and will clarify this more explicitly in the manuscript. While this work is a technical proof of concept for the retrieval of high-resolution temperatures, applied to a single test site under clear-sky and, low-wind conditions, we have also clearly and quantitatively shown (using a correlation of over 6000000 data points) in what is considered a relatively large coverage and high-resolution, of local in-situ surface temperatures over snow on sea ice, that microtopography plays a dominant role for local surface temperatures under clear sky conditions. It is also clear that this would not be the case under diffuse conditions, because of the lack of topography dependent irradiance. We will emphasize the last point in the manuscript.

The use of the TIR camera is restricted to clear-sky periods, as clouds would force the surface temperature to reach thermal equilibrium with the cloud base temperature, overruling the impact of snow/ice spatial heterogeneity on the surface temperature map. While we recognize the value of expanding the analysis to additional sites or varying environmental conditions, such extensions are planned for future work and lie beyond the scope of this study.

4. Additionally, the snow depth proxy derived from the UAV DEM rests on the assumption of a flat ice surface and uses a single calibration offset from ground-based measurements. But where is the validation? There is no high-precision GNSS positioning of the MagnaProbe data, and the ±0.05 m uncertainty in snow depth is not propagated into any of the correlation analyses or surface temperature interpretations. How does this uncertainty affect the reliability of the snow temperature correlation results? Are there areas where DEM uncertainty dominates the signal? This aspect of the study is underdeveloped and undermines confidence in one of its core data layers.

AC: We agree that the snow depth proxy represents a potential source of uncertainty in our analysis. The approach relies on the assumption of a flat ice surface and a single offset calibration based on 813 ground-based magnaprobe measurement points, which do not

have high-precision GNSS positioning. We will add a discussion on the estimated ±0.05 m uncertainty in the snow depth proxy and its potential impact on the snow temperature correlations. While this level of uncertainty is consistent with previous UAV-based snow depth studies, we recognize that error propagation has not been explicitly addressed and will incorporate this into the revised manuscript to better qualify the reliability of the snow depth-temperature relationships.

5. A further issue is the lack of an overall uncertainty framework. The authors had ±2.1°C for temperature correction (Figure 4), but there is little analysis of how this uncertainty varies spatially (e.g., near image edges, in shaded areas), or how it might interact with surface roughness, sediment detection, or irradiance calculation. What are the sources of uncertainty from lens effects, and how do they affect the final interpretation? Could they lead to systematic biases in warmer sediment zones or sloped surfaces?

AC: We did not clarify this sufficiently in the original manuscript, but we did not use the entire image for our calculations, instead, as explained in the workflow figure 3, "Oval image mask applied to account for the vignette effect (corners of the image are cooler than the centre) in thermal images due to the lens". So, the image edges were actually removed and not included in the dataset. We will rephrase "account for" to "remove the areas affected by" to be precise. We will attempt at a clearer and more comprehensive discussion of uncertainties.

6. There is also a surprising absence of any actual surface energy budget analysis. The paper frequently references "energy balance" but never attempts to break it down into its components (e.g., incoming solar, reflected solar, outgoing longwave, conduction). Without this, the proposed mechanisms, e.g., sediment warming or irradiance enhancement on slopes, remain less clearly. Why not use the DEM and albedo proxies to estimate slope-corrected irradiance per pixel, or combine reflectance and temperature into a basic radiative balance estimate?

AC: Considering that we have calculated the slope-corrected irradiance and we discuss it in quite a bit of detail (e.g. section 2.3.9, figure 10, section 3.2.3., figure 12, figure 13, figure 14, section 4.4), we find this comment a bit puzzling. Our discussion about the slope corrected irradiance and its correlation with the measured surface temperature seems to be exactly what the reviewer is missing, so we are not quite sure how to address this comment. We do not have albedo proxies, as our RGB measurements are not set up to retrieve albedo. Conducting a comprehensive energy balance breakdown is beyond the scope of this study, which is focused primarily on mapping spatial variability in snow surface temperature and exploring its potential drivers. Additionally, EB modelling would require a lot of sensitivity analysis to input parameters and boundary conditions, to account for the large variation and uncertainties in local EB input parameters and boundary local conditions. We will revise the manuscript to better frame the discussion around temperature variability and avoid overstating conclusions related to the full energy budget.

7. Finally, the overall discussion remains descriptive and lacks synthesis. What is the key physical message here? Are the findings consistent with prior studies in the Antarctic? What is new about the relationship between microtopography, sediment, and surface temperature, and what does it imply for satellite-based retrievals or sea ice models? Without clearer integration of results into a broader scientific context, the paper risks reading more like a high-resolution mapping exercise than a process-oriented scientific study.

AC: We appreciate the reviewer's insightful comment and agree that a clearer synthesis is needed. We will revise the discussion to better highlight the key physical insights and explicitly connect our findings to the limited existing literature and data on Antarctic snow on sea ice, which remains scarce. This scarcity is a primary motivation for our study. We will emphasize what is new about the observed relationships between microtopography and surface temperature and discuss their implications for satellite retrievals and sea ice models. The relationship between surface temperature and sediment is relatively well established and gives our results significant confidence because we can clearly observe it. Our aim is, as we have done here, to move beyond descriptive mapping and provide quantitative statistics and process-level understanding relevant to this underexplored region.

Specific comments:

1. Section 2.3.3: The snow depth proxy construction is hard to follow. Are you aligning in situ snow depth data with the DEM by GPS coordinates, or using a 3 stereophotogrammetric offset? Please clarify the procedure with a schematic or simple diagram. This section needs better structure to explain how the reference elevation is selected and how uncertainty is estimated.
AC: We will clarify the procedure in Section 2.3.3 by clearly describing how the in situ snow depth measurements are aligned with the DEM using GPS coordinates. Additionally, we will explain how the reference elevation is selected, and detail how the associated uncertainties are estimated to improve the section's clarity and structure.

2. Figure 8: (1) The latitude/longitude labels are unconventional and difficult to interpret; (2) The color bar is unintuitive (blue implies deep snow, which is opposite of expectations). Consider reversing the color scale or annotating it more clearly. (3) Does the 0 m value mean bare ice? Please explain this in the caption.
AC: We will revise Figure 8 by correcting the latitude and longitude labels to a more conventional format for easier interpretation. Additionally, we will adjust the color bar to better reflect intuitive. The figure caption will also be updated to clarify that the 0 m value corresponds to bare ice areas.

3. Figures 10 and 13. These are not well-integrated into the discussion and don't provide substantial new insight. Consider removing them or clearly stating their purpose.

AC: We will add more explanation for Figure 10 to clarify its relevance, including details on aspect and its role in the analysis. Regarding Figure 13, we are considering removing it as it does not provide substantial new insight and is not well integrated into the discussion.

4. Line 300:"…to/by 0.53 °C?". Please clarify whether this value is a change "by" or an absolute value "of".
AC: We acknowledge this comment, which is similar to one raised by Reviewer 1. We are going to clarify that the RMSE for surface temperature decreased by 0.53 °C, down to 0.58 °C.

5. Line 486: "…area of 200 m.". This is not an area. You probably mean "length of 200 m" or "area of 200 m × 200 m".
AC: We agree and will correct the phrase to "area of 200 m × 200 m" in the revised manuscript.

6. Line 332 and Line 568: remove one "the"
AC: We will remove the redundant "the" in both instances.

7. Figure A2 and A3 should be swapped.
AC: We will swap Figures A2 and A3 as suggested.

8. Figure A4 (b) and (c) add little value, it's redundant with (a). Consider removing or merging.
AC: We are going to improve this Figure.

---

## Author Comment (AC3)

**03 Review**

General

This article examines the spatial variations in snow surface temperature at the plot scale (hundreds of meters) on sea ice, aiming to identify the primary drivers. Snow temperature is highly sensitive to even minor changes in incoming fluxes (such as shortwave and longwave radiation) and serves as a means to explore the surface energy budget (SEB) and its unique characteristics on snow. To address this question effectively, the authors collected a unique and exceptional dataset in the Antarctic. The paper holds significant interest for the TC community but has some shortcomings, one of which is notable yet addressable.

The main issue lies in how the paper's question and objective are approached. The hypothesis proposed by the authors to justify and structure the study is unconvincing from my perspective, and they ultimately demonstrate that this hypothesis is indeed not validated. Instead, they identify one/two more adequate hypotheses, which, to my knowledge, are more obvious. While it is appreciable to build a paper around a clearly stated hypothesis, the fact that this hypothesis is not commonly supported in the literature and can be dismissed with magnitude order calculations undermines the paper's construction. What if the hypothesis had been, "Does local topography explain most of the surface temperature variations at the plot scale?" Research on the link between topography and SEB is prevalent in the literature, and the current title aligns more closely with this hypothesis. With the same dataset and results, the introduction and discussion would flow better, aligning more effectively with the results and conclusion.

Apart from this shortcoming, the paper is excellent and has many strengths, including significant work on TIR camera correction and an exceptional dataset. The paper is clear and well-written, with an easy-to-follow logic (except the order of the figures) and a generally sufficient level of detail (see the exception on the model below). However, the abstract is difficult to understand without having read the paper, as the logical progression is unclear (unlike in the paper itself). To attract more readers, I suggest rewriting the abstract from scratch, incorporating more results and fewer hypotheses.

The paper is quite long and my recommendation for the review is to shorten where possible, but avoid lengthening (except for the model description!).

AC: We appreciate the reviewer's thoughtful comments and fully agree with the concerns raised regarding the framing of our hypothesis. We will revise the manuscript to shift away from the narrow focus on snow depth as the primary driver of snow surface temperature

variability. Instead, we will adopt a more comprehensive and better-supported hypothesis, aligned with our findings: *"At the plot scale on Antarctic sea ice, spatial variability in snow surface temperature is primarily driven by local surface conditions, including microtopography, sediment presence, and variations in irradiance, rather than snow depth alone."*

This re-framing will be reflected in a revised introduction, clearer discussion, and an updated abstract that emphasizes our key findings rather than the initial hypothesis. We agree that this restructuring will improve the logical flow and impact of the paper.

We also acknowledge the comment regarding manuscript length and will make targeted reductions to enhance clarity without compromising content. Additionally, we will expand the model description section to ensure sufficient detail is provided for reproducibility and transparency.

**Detailed comments:**

L 6 "Our airborne maps reveal a mean snow depth of 0.16 ± 0.06 m". The mention of snow depth measurements is new in this sentence. The previous sentence is about surface temperature and the end of this sentence is about surface temperature. A reorganisation is necessary.

AC: We agree and will revise the sentence to improve clarity and logical flow by separating the introduction of snow depth measurements from the discussion of surface temperature. This reorganization will help ensure that each concept is introduced in a more coherent and contextual manner.

L11 "seemingly flat snow field" can you give a number, this statement is relative to reader's expectation of what flat is.

AC: We agree and will quantify the flatness of the snow field by including the observed range in snow depth, as well as provide a description of the overall roughness features (absence of pressure ridges and rafted ice) to clarify what we mean by "seemingly flat" and make this statement more objective.

L13-14. It is not clear what the variability accounted in the uniform irradiance model.

What is meant exactly by "the incoming solar radiation (irradiance) at the point scale". In principle incoming solar radiation is measured w/r to a horizontal surface. Do you mean the solar radiation received / perceived by the surface ?

AC: We thank the reviewer for pointing this out. To clarify, our irradiance model computes the solar radiation actually received by each inclined sea-ice surface element, accounting for local slope, aspect, solar geometry, and terrain shading—rather than using irradiance

values referenced to a horizontal surface. It combines measured global and diffuse radiation during the flight with the DEM-derived surface geometry using established radiative transfer approaches. We will revise the manuscript to better explain this and eliminate ambiguous phrasing such as "incoming solar radiation at the point scale."

L17 "While we initially hypothesized that snow depth was a key driver of snow surface temperature," this hypothesis should be stated in the first part of the abstract (L5) to position the problem addressed in the paper, and it should be justified, backed by literature because this hypothesis is not intuitive, at least to me. From the SEB equation, I'd expect surface temperature to depend on irradiance first, the snow depth does not appear in this equation unless the conductive term is written with the Fourier law. However even in this case, the snow is far too insulating to allow this term to become significant with respect to the others in summer, especially the incoming irradiance.

AC: Our original hypothesis was motivated by the fact that, because of the strong insulative and scattering properties of snow, in case of shallow snow covers small changes in snow depth have a strong impact on the heat conduction and surface albedo. How much this affects the surface temperature will, however, very much depend on ice thickness, air temperature, and range of albedo variability: surface temperature is more sensitive to snow depth for thinner ice, colder air temperature, and large albedo contrast between bare ice and snow. In this case study, the very thick sea-ice (2.4 m) strongly insulated the snow layer from the ocean heat flux and also caused a relatively large bare ice albedo. Both these effects minimized the sensitivity of surface temperature to snow depth. We will revise the abstract and introduction to frame our initial hypothesis more broadly - acknowledging that while we considered snow depth as one possible contributor to spatial variability in surface temperature, we also evaluated other controls, particularly irradiance and topography. We will support this revised framing with references to relevant literature on energy balance partitioning over snow-covered sea ice. However, we would like to point out than when solving the equations for the SEB, snow depth a) plays a role in the heat conduction, and b) it influences the lower boundary condition, which is the snow-sea ice interface, so snow depth does certainly play a role for the surface energy balance. This may not be relevant for thicker snowpacks, but for the relatively thin snowpacks that are observed in regions of Antarctica and the Arctic, the snow depth (and the snowpack's physical properties) has a dominant role in the heat exchange in the air-ice-ocean system.

L34-35: "through the satellite period" "since record-keeping began". Should indicate the starting year to avoid ambiguity.

AC: Agreed. We will specify the starting year for both the satellite observational period and historical record-keeping to ensure clarity and remove ambiguity for the reader.

L47. While the authors are free to make the hypothesis they want as long as it is clearly stated – and I acknowledge it is very well done here w/r to the literature in general -- still I found this hypothesis strange. The physical reasoning behind this hypothesis should be developed a bit and examples from the literature could help.

AC: We appreciate the reviewer's feedback. While we initially presented the hypothesis focused on snow depth, we agree that incorporating surface topography and irradiance provides a more physically grounded framework. We will revise the hypothesis accordingly and include relevant literature examples to better support the physical reasoning.

L90. The objective would benefit to be rewritten without this simple hypothesis. A more neutral approach would be to list all the potential factors influencing the small scale variability of snow surface temperature with a short literature review for each, and reframe the goal into investigating/quantifying which term is the key driver in the specific context of this study (summer, sea-ice with small snow depth).

AC: We appreciate the reviewer's insightful suggestion. We will revise the objective to present a more neutral approach by outlining the potential factors influencing small-scale variability in snow surface temperature, supported by a brief literature review. The objective will then focus on investigating and quantifying the dominant drivers within the specific context of summer sea ice with shallow snow depths.

L116: "The other four sites" I'd remove this sentence, it diverts from the objective of the paper.

AC: We agree that this sentence somewhat diverts from the main objective of the paper. However, we will retain a brief mention to provide context on how this study fits into our broader ongoing research, which will extend these findings to additional sites.

L136. Please check the correspondence between he height of installation vs the footprint areas (1.6m2 versus 0.35m2). Is the angle of installation at the sediment site different ?

AC: We confirm that the angle of installation at the sediment site differs from the other sites, which results in a different footprint area. We will clarify this point in the manuscript for better understanding.

L214: Check that the figures are referred in order (it seems Fig 8 is referred before others).

AC: We agree and will ensure that all figures are referred to in the correct order throughout the manuscript.

L215 L223: Check figure A1 reference

AC: We agree and will verify and correct the references to Figure A1 to ensure they are accurate and properly placed.

Figure 2. Can you show a scatterplot (+ r2 and RMSE) between the inferred snow depth proxy and the magnaprobe measurements as a validation of the approach ?
AC: AC: Thank you for the suggestion. We initially considered a direct point-by-point comparison between the inferred snow depth proxy and the magnaprobe measurements, but due to the known horizontal uncertainty in the magnaprobe's GPS (up to several meters), we found that such a comparison would be misleading, especially given the high spatial variability of snow depth at our site. A small spatial offset can result in large apparent discrepancies. For this reason, we chose a histogram-based comparison, which provides a more robust and meaningful assessment of the proxy's ability to capture the distribution of snow depth.
To further support this, we will include an illustrative example showing the potential variation in measured snow depth within a radius that corresponds to the GPS uncertainty (e.g., 3–5 m), highlighting how much depth variability can exist within a single GPS error footprint. We believe this better reflects the true agreement between the datasets than a scatterplot would.

L272: It is not obvious how 0.5 °C was found based on Fig 4a. I guess it is empirical but how sensitive to conditions is it?
AC: We agree that this was not sufficiently explained. The 0.5 °C threshold was determined empirically based on the stability of the NUC correction curve observed in Fig. 4a. We will clarify this in the text and add a brief comment on the sensitivity of this threshold to varying environmental conditions.

L318: by curiosity, how large is this correction ?
AC: We interpret the reviewer's question as referring to the magnitude of the correction between the Apogee brightness temperature and the derived surface temperature in Eq. 4. For the Apogee pointed at the sedimented snow surface, the mean correction is 0.55 °C.

L321: I don't understand what this RMSE is, between what and what it is calculated (+ typo: with an RMSE => with a RMSE)
AC: Thank you for the comment. This RMSE refers to the temperature anomaly RMSE calculated after applying the NUC (non-uniformity correction). It quantifies the residual error in the temperature dataset and is used as the uncertainty estimate for the surface temperature maps. We will clarify this in the manuscript and correct the typo.

L329: " The RMSE of the residuals of this linear fit". RMSE → RMS or remove residuals
AC: Going to change the phrasing.

L330. "the square of the thermal" and "the square of the RMSE associated " check if square is correct in both cases. I'd recommend to completely rephrase or write as an equation.

AC: We thank the reviewer for pointing this out. We agree that the phrase "RMSE of the residuals" is redundant, as RMSE is inherently calculated from the residuals. We will revise the sentence to read "The RMSE of this linear fit" for clarity.

Section 2.3.8. It is not clear how the impurities are detected. Using a threshold or just visually ? If just visually, this section could be removed, and a line or two in the results section is sufficient.
AC: We agree with the reviewer. Since impurity detection was done visually, we will remove Section 2.3.8 and instead include a concise description of this step in the results section where the findings are discussed.

Section 2.3.9. Given the critical role of the model, the level of detail should match that given for the drone. It is necessary to provide the main aspect of the model (e.g. workflow) and present the equations or refer to the equations in the cited papers for each main calculation step.
Main questions are: the diffuse component, the resolution of the calculation (in relation with the positioning uncertainties), cast shadows, multiple scattering (esp for the shadows).
AC: We agree that additional detail on the irradiance model is warranted. We will revise Section 2.3.9 to include a clearer description of the model workflow and specify the treatment of the diffuse component, calculation resolution, shadows, and the handling (or limitations) regarding multiple scattering (there is no multiple scattering, and the diffuse component is just the measurement recorded by the SPN1 sensor), particularly in shadowed areas (which are a very small percentage). We will also add relevant equations or explicitly refer to the corresponding formulations in the cited literature.

L360. How does 2.4 ± 0.04 m translate into 1% variation and where 0.04m is coming from ? The histogram seems to indicate larger deviation. How relative variation is defined and calculated ?
AC: We thank the reviewer for the comment. The value of ±0.04 m refers to the standard deviation of all 2449 sea ice thickness measurements and was not used for calculating the relative variation. The 1% variation was derived from the mean absolute deviation (MAD) of the thickness values, expressed as a percentage of the mean thickness (2.4 m). We will revise the text to clearly distinguish between standard deviation and MAD and report the correct value used for the relative variation.

L395. Fig 12 is referred, check the order.
AC: Thank you for pointing this out. We will ensure that Figure 12 is referred to in the correct order within the manuscript.

L413. Isn't it due to the correlation between snow depth and impurities ?
AC: We thank the reviewer for the suggestion. However, we do not think the stronger correlation is due to a direct relationship between sediment and snow depth. The "sediment" dataset includes outlines drawn around sediment patches, but these masks

also include surrounding clean ice pixels, introducing greater variability in both snow depth and temperature. In contrast, the "no sediment" dataset is based on rectangular boxes placed in uniform snow areas, resulting in more homogeneous conditions and a narrower temperature range. Therefore, we attribute the stronger correlation in the "sediment" dataset to this broader variability rather than a direct link between sediment cover and snow depth.

L469. Why is this relevant in this section ? It is well known that the irradiance depends on the local incidence angle, and not on the slope.
AC: Thank you for pointing this out. We agree that irradiance strictly depends on the local incidence angle, not slope alone. However, in this section, we refer to slope as a proxy for estimating how the local surface orientation influences the irradiance received - i.e.," tilted irradiance". We acknowledge that the term "irradiance" may have been used imprecisely here, and we will clarify the language to reflect that we are referring to slope-driven variations in local incidence angle, which in turn affect the effective irradiance received by the surface.

L476: The aspect distribution is uniform, not gaussian. The slope distribution is not Gaussian.
AC: We agree and will correct the statement.

Figure 12. For convenience, adding titles on the rows directly in the graph would help quickly read the figure, without having to read the caption.
The x-axis scale is very large, for just a few outliers. I suggest to reduce the range to -17°C - -10°C or so. It would make the graphs H and I more convincing for instance.
AC: We will add titles to each row directly within the figure to improve readability. We also agree with the suggestion to limit the x-axis range to reduce the influence of outliers and better highlight the distribution patterns, particularly in panels h) and i).

L491: I'm not sure but I think that the algorithm not only correct for NUC jumps but also for other trends in the camera which are usually very large.

Note that other cameras do a "better" job in NUC smoothing which makes jumps more difficult to detect... while still be necessary to applied the necessary corrections. See for instance: Arioli, S., Picard, G., Arnaud, L., Gascoin, S., Alonso-González, E., Poizat, M., and Irvine, M.: Time series of alpine snow surface radiative temperature maps from high precision thermal infrared imaging, Earth Syst. Sci. Data, 16, 3913–3934, doi: 10.5194/essd-16-3913-2024, 2024

Ideally, one would access the raw data... but it seems that camera manufacturer prefer to overprotect their (insufficient) algorithms.
AC: Thank you for pointing this out and for the great reference. We agree that our correction algorithm is relatively simple and may not capture all forms of sensor drift or be

directly transferable to all camera models, especially those with more complex or opaque NUC behavior. Still, we see it as a useful baseline that demonstrates the value of transparent correction methods and offers a starting point for adaptation in other environments and systems. Importantly, this correction was necessary to make the thermal data usable at all - without it, none of the images could have been processed reliably.

L500. I'd advocate for m ore accurate sensors than Apogee sensors when absolute value is important (e.g. close to 0°C, see Arioli et al. 2024)
AC: We agree with this statement. A colleague of ours has developed a really nice method of large metal plates (painted with a black paint of known emissivity) on top of an insulative foam and a thermocouple that logs the temperature of the plate. We plan to use that in the future.

L539. "While the red band values do not directly affect the surface energy balance, we use them as a proxy for impurities." I don't understand this statement. Maybe the verb "affect" is incorrect.
AC: We agree the original phrasing was unclear. We will revise the sentence to clarify that while the red band values alone cannot be used  to infer surface impurities for surface energy balance calculations (and albedo), in our case, they serve as a proxy for surface impurities (and potential energy absorption by the impurities). But we acknowledge that the red band values are influenced by illumination as well by impurities, as discussed throughout the manuscript.

L550. I would suggest to coarsen the resolution a bit to account for the positioning uncertainty and to see how this correlation increases. Mathematically, the correlation always increases with smoothing, but here the idea is to see how quick it increases.
AC: We acknowledge this is an interesting suggestion. However, performing the smoothing analysis would require substantial computational time and resources, which we cannot accommodate within the current revision timeline. We will consider this for future work.

L567: the the => the
AC: We will correct this typo.

L570: while this result is sound, the statistical demonstration would require first to demonstrate that topography and impurities are independent in your case. For instance if the impurities areas had more north looking slopes, the relationship is biased. It is frequent (in the mountains) that the sun facing slopes are more likely to have dust emerging at the surface than the colder faces.
AC: We appreciate the comment. However, in our sea ice setting, sediment is likely wind-deposited rather than emerging from the snowpack due to melt, as is common in mountainous terrain. That said, we do observe some correlation between sediment presence and topography, with sediment-covered areas tending to be higher and in areas

of thicker snowcover. We will clarify this relationship in the manuscript.

L580 I suggest to also mention multiple scattering which is likely important in the cast shadows areas and cavity effects in the LW which is probably negligible with slopes <10°. Ref: A. Robledano, G. Picard, L. Arnaud, F. Larue, I. Ollivier, Modelling surface temperature and radiation budget of snow-covered complex terrains, The Cryosphere, 16, 559–579, doi:10.5194/tc-16-559-2022, 2022

AC: We will mention the potential role of multiple scattering in cast shadow areas but note that such shadowed regions constitute a very small fraction of the study area (less than 1%), limiting the overall impact. We will also acknowledge that cavity effects in longwave radiation are likely negligible given the generally low slopes (<10°) observed in our site. Additionally, the small-scale multiple scattering also affects the albedo, which we have already discussed in the manuscript.

L608: This is not necessarily a drawback. If only the irradiance is changing (not Tair, not wind), observing two different Ts give a lot of information on the balance between SW and the other terms of the SEB.

AC: We thank the reviewer for this insightful comment. The two flights were necessary to acquire the RGB data for constructing the DEM, and we combined the thermal infrared data from both flights to generate a single, averaged orthomosaic for this study. For future work, we are considering analyzing the flights separately or using just one flight to build the orthomosaic, which could allow investigation of surface temperature changes related to varying irradiance.

L630: "offering valuable tools for many users". Same comment as before. To my experience, using more expensive cameras with better NUC correction make the proposed solution not applicable and the problem more severe... A recommendation could be to buy or develop open-source cameras or at least cameras that have been evaluated by others and for which correction algorithms exist.

This is a very good point, and we appreciate the reviewer pointing out that better cameras may make this problem even worse. So, we are pleased to provide a solution to this relatively "inexpensive" off-the-shelf DJI setup, which has proven to reliably work in polar regions. The heated DJI batteries used with our camera enable sufficient flight durations to cover our test fields (200 × 200 m), which remains challenging with non-heated alternatives. Many custom or alternative cameras face limitations such as insufficient battery life under the low temperatures typical of Antarctic conditions, but it's a good idea for the community to keep looking for open-source alternatives. We will add a note discussing these practical considerations in the revised manuscript.

---

## Author Response (AR1)

**Dear Editorial Team,**

We sincerely thank you for the time and effort dedicated to coordinating the review process for our manuscript. We greatly appreciate the reviewers' constructive comments and suggestions, which have helped us to substantially improve and restructure the paper. In this document, we provide detailed responses to all three reviews, with the corresponding revisions highlighted in the accompanying tracked changes PDF file.

Sincerely, Julia Martin

**01 Review**

**General**

The authors investigate how snow distribution patterns affect the surface temperature of snow on Antarctic sea ice, a key but under-studied factor in the sea-ice energy balance. The introduction puts the topic of the paper well into context and provides a great overview of the current state-of-the-art. To contribute to this vast topic of polar sea ice, the authors applied UAV and ground measurements to create high-resolution maps of snow topography and surface temperature over uniform landfast sea ice in McMurdo Sound, Antarctica. The measurement site, the ground-based and airborne methods are introduced in detail. Regarding the latter, a novel algorithm was developed to correct thermal drift in UAV thermal imagery, ensuring consistent temperature data. Based on these maps and correlations, the authors investigated the reasons for observed surface temperature variations. As a result, the surface temperature anomalies were mainly linked to visible sediment on the snow, not snow depth, which has been the authors initial hypothesis. They further found that small-scale topography significantly affected local solar irradiance, and assuming uniform irradiance underestimated its variability. Overall, sediment and irradiance were found to have a stronger influence on snow surface temperature than snow depth, highlighting the importance of surface features in energy balance modelling on sea ice. The manuscript provides an important contribution to the analysis of polar surface temperature variations. It presents interesting and valuable results, which help to identify gaps in common surface temperature retrievals assuming flat surfaces. It highlights the problems and mismatches they struggle with and introduces proper solutions. I highly recommend its publication after the authors have revised the manuscript regarding the comments listed below.

**Major comment**

**Length:** The paper is very long, which makes it difficult to keep the readers attention from the beginning to the end. However, I think there is some potential to shorten the paper significantly.

1. In my opinion, there is no reason to separate the results and discussion sections. On the contrary, the separation results in a lot of repetition, which unnecessarily lengthens the paper. By merging the two sections, this could be avoided, and the paper could be significantly shortened, which would also

make it more focused. The individual sections already have similar topics, so this should be easy to do.

L367: We merged the Results and Discussion section restructured the section in

- L368: 3.1 Airborne maps: red band orthomosaic, snow depth proxy and surface
- L394: 3.2 Possible drivers of small-scale surface temperature variations: snow depth, sediment deposition or irradiance?
  - L399: 3.2.1 The role of snow depth
  - L430: 3.2.2 The role of "darker" areas: sediment or shadows?
  - L478: 3.2.3 The role of irradiance
  - L527: 3.2.4 Integrating predictors via linear mixed-effects modeling
  - L543: 3.2.5 Technical considerations for high-resolution mapping of snow topography and snow surface temperature
- 2. There are several graphs that represent more or less the same thing. For example, Fig. 1a,b and Fig. 5 a,b. One figure, either a or b, would be sufficient here. Some figures can be merged. For example Figure 8 and 9. Why not have one column with the Red Band Value (G and B are not really used), one column for the DEM and one for the temperature? Furthermore, there are graphs that in my opinion are not needed at all. For example, Figures 10 and 13. I have also made additional suggestions under Minor and Technical Corrections.
- Fig. 1: We expanded Fig. 1b and deleted Fid. 1b
- Fig. 5: We deleted Fig. 5b and merged Fig. 5a with Fig. 4 (now Fig. 3).
- Fig. 8, 9: We merged these into the new Fig. 7, with 3 columns as suggested by the reviewer: a) the red band orthomosaic, b) the DEM and c) the TIR orthomosaic. We deleted the RGB orthomosaic. We also changed the x- and y-axis to geographical coordinates to improve readability.
- Fig. 10: We kept Fig. 10 (now Fig. 11).
- Fig. 13: Deleted.
  - 3. Some further sections can be skipped as own sections and merged into others. For example, the main message from 2.3.4 fits to the introduction of the camera, Section 4.5 belongs to the summary and conclusion part.

We restructured the Methods section and merged the sections as suggested by the reviewer.

- L169: 2.3 Airborne data
  - L181: Georeferencing
  - L190: 2.3.2 TIR image processing: Non-Uniformity Correction (NUC)
  - L261: 2.3.3 TIR orthomosaic
  - L272: 2.3.4 TIR orthomosaic: Ground-based temperature calibration
  - L296: 2.3.5 Digital Elevation Model (DEM)
  - L310: 2.3.6 Snow depth proxy

L345: 2.3.7 Irradiance calculation

We deleted the section 2.3.8 Detection of sediment deposition and incorporated it into Section 3.1 to reduce repetition.

**Figures:** The figures are often not really introduced, but are only mentioned in brackets after certain statements, so that the reader has to find out for himself what is shown. This makes it difficult to read fluently and understand directly. It would be good to describe what is shown in the text with one or two sentences.

In the final stage we thank the reviewer for this suggestion but decided to not follow it. Even though we see the benefit in terms of improved flow of reading, we think that the figures are sufficiently explained in the captions and hence this becomes a style question. We prefer to refer to the figures in brackets rather than explaining the content in the actual text as this would contribute to lengthening the paper.

**Minor comments:**

1. Order: Figures are partly not numbered in the order they are used in the text.

We reordered the figures to match the order they are mentioned in the manuscript.

2. Space signs: Space signs are multiple times missing between numbers and units, between figure abbreviation and figure number and in front of citations that are given in brackets.

We fixed all spacing inconsistencies throughout the manuscript, including those between numbers and units, figure abbreviations and figure numbers, as well as spacing before in-text citations.

3. Indices and units: Indices are sometimes written in italic letters and sometimes in non-italic letters. For reasons of consistency, you should write all indices in non-italic letters.

We standardized the formatting of all indices in the manuscript; they are now written in non-italic letters for consistency.

4. P2, L46: What means the original hypothesis was? It should stay the same, although it might have been rejected.

We changed the sentence. L40: "Despite the recognized influence of snow properties on sea ice thermodynamics, the relative importance of different factors - such as snow depth, surface microtopography, and sediment deposition - on snow surface temperature remains poorly understood."

5. Sect. 2.3.2: You might have to revise this section to make it more clear to the reader what happens here. I had to read it several times and I am still not sure if I got it right. Do you match the in situ snow depth measurements by their GPS position into the RGB images? Or is it some kind of a stereographic method? Please revise it to make this easier understandable. Maybe also a sketch might help to better understand the procedure behind.

We rephrased section 2.3.3. to clearly state that we added a uniform offset of 0.16 m - the difference between the shallowest measured snow depth and the lowest DEM elevation - to the entire detrended DEM to approximate snow depth for each pixel. This adjustment produced a proxy snow depth map, with a vertical uncertainty of  $\pm 0.045$  m, based on propagated DEM and MagnaProbe errors. We didn't include a (new) schematic as in our opinion, Fig. 5a and Fig 5b show the method sufficiently. We

explained why no direct comparison between MagnaProbe snow depth and DEM elevation values was possible (because of the high uncertainties in the MagnaProbe GNSS positions). We also show this in Fig. A3 by comparing six randomly chosen MagnaProbe measurements against all elevation (dark gray) and snow depth proxy (light grey) values derived from the DEM within the 2.5 m radius around these six MagnaProbe locations. These plots show that the variability within a single GPS error footprint is almost as high as the variability across the whole measurement field, demonstrating why a direct scatterplot comparison would be misleading.

6. Sect. 2.3.4: I don't think that this section is really needed. It should be enough, if you mention the conversion within one sentence, when you introduce the camera.

We merged section 2.3.4 with Section 2.3.5, as suggested above, and condensed the content to a brief mention of the conversion before introducing the camera: L191 – L193: The proprietary RJPG format used by the thermal camera retains absolute brightness temperatures but cannot be directly analyzed. We therefore converted the images to TIFF using the ImageJ IR UAV plugin (Pereyra Irujo, 2022) before processing them with the SfM workflow in Agisoft Metashape Pro version 1.8.4 (Wigmore and Molotch, 2023) in Fig. 2.).

7. Sect. 2.3.5: This is a very well-thought-out method, which leads to convincing results. I just wonder how you derive the absolute calibration. Is it a predefined function for each pixel from the lab, which is then scaled for each pixel by the NUC? Or do you know the temperature of the shutter (which then acts like a black body) and in parallel is used as a homogeneous target to remove the non-uniformity of the single pixels? Furthermore, later in Line 305 you discuss a vignette effect, which originates from the lens properties. That brings me to the question where the shutter is installed. Is it installed in front of the lens or between the lens and the detector? For the latter you will imprint the structure of the lens-own temperature into the images, when you perform the NUC, which then leads to this vignette effect and might also changes the absolute calibration.

Thank you for the insightful questions. Unfortunately, the camera's internal calibration process is not fully accessible to us, and detailed information on the absolute calibration or shutter placement is not provided by the manufacturer (investigation with DJI was unfortunately unfruitful). The vignette effect is taken care of by applying an oval mask as part of the AGISOFT Metashape processing workflow shown in Fig. 3. We added a sentence to clarify that the vignette correction was done by applying an oval-shaped mask (L263).

- 8. Sect. 3: The initial paragraph is a repetition and can be skipped. We removed the initial paragraph from Section 3 to avoid repetition.
  - 9. P24, Fig 12: It would be great to have such a "correlation" plot for "degree of sedimentation" vs. temperature. I guess here you would find a significant high correlation. Of course, I see the point with the difficulties related to the varying camera settings as you write in Line 337 and the shadows you mention in Line 408. But shouldn't be the first issue solved by your NUC calibration? Furthermore, you could use your DEM to extract regiones, which might be affected by shadows. Adapting both, you would be able to select scenes, which are unaffected by shadows and after a normalization you can

extract an arbitrary value for the degree of sedimentation, which might range from 0 to 1.

We appreciate the suggestion, and the level of detail that the reviewer has thought about this; however, the 'degree of sedimentation' is derived from RGB imagery, which is not corrected by the thermal NUC calibration. Since RGB values are affected by varying illumination and camera auto-settings, they do not provide consistent surface reflectance measurements, and even if we remove the totally shaded areas, the issue of the varying illumination remains. Therefore, developing a rigorous, normalized sedimentation index from these data is currently not feasible within the scope of this study.

10. Sect. 4.5: Belongs to summary and conclusions. There, it fits well at the very end.

L565: We moved the paragraph into section 3.2.5 Technical considerations for high-resolution mapping of snow topography and snow surface temperature. There it fits well with the summary of technical considerations and limitations.

11. Sect. 5: I would expect some words regarding the correlations and the main findings resulting from them.

We included a summary of our main findings in section 4 Summary and Conclusions. L583: Our high-resolution data reveal how small-scale spatial variability in snow surface topography strongly influences irradiance and surface temperatures. Assuming a flat ice surface underestimates irradiance variability, potentially affecting non-linear energy balance processes such as localized melting and sediment transport. We found that irradiance and surface impurities dominate surface temperature patterns, while snow depth plays a secondary role.

**Technical comments**

We addressed all points raised by the reviewer.

- Point 17: We appreciate the reviewer's attention to detail. We have carefully considered this and note that temperature differences are consistently reported in degrees Celsius throughout the manuscript for clarity and consistency.
- Point 19: We clarified that the RMSE for surface temperature decreased by 0.53 °C, down to 0.58 °C".
- Point 28 (Fig. 10): We believe this figure is valuable as it shows the range and distribution of calculated irradiance values across the study area. While aspect itself may not be a primary focus, it provides context about solar exposure variability. We reordered the figures and believe that Fig. 10 is now incorporated sufficiently.
- 1. Sun: "Sun" is a proper name and should be written with capital letter. It appears several times throughout the manuscript.

  Done.
- 2. **P1, L13:** ... our study (200x200 m) area  $\rightarrow$  ... our study area (200 x 200 m2) L4: We changed the phrasing to "200 m x 200 m".
  - 3. **P2, L26:** Wendisch et al. 2023 might be cited as well as it summarizes a huge project with its focus on Arctic amplification.

Added reference in L20.

4. **P2, L53:** → ...(), colder

Fixed.

5. **P3, L78:** Acronym

Fixed in L68: Polar Radiant Energy in the Far Infrared Experiment (PREFIRE)

6. **P3**, **L85**: Which infrared? Near-, thermal-, far-?

Fixed in L74: (visible, thermal infrared).

7. **P4, Fig. 1:** Only Fig. 1a is needed. Figure 1b does not offer to many more details. If you prefer to keep both, add E(ast) and S(outh) in Fig. 1b and draw (a) and (b) into the Figures. I have overseen it for a long time. Furthermore, there is a missing space sign in the figure caption before (yellow rectangle).

We changed the figure to only consisting of 1 panel (Fig. 1a) and deleted the redundant one (Fig. 1b).

8. P4, L92: Add an outline at the end of the introduction?

We added a short paragraph outlining the scope of the paper (L83).

9. **P5, L115:** Size and time period are already mentioned in the sentence before. Skip the repetition.

Done. Deleted time period and size.

10. **P5, L129:** Italic index, which might be a typo. Also, two lines later. Fixed.

11. **P6, Table 1, caption:** Skip "used in this paper". Should be clear. Done.

12. **P6, Table 1:** Maybe include some empty lines in between the different parameters. Since some parameters use two lines it is hard to see, which entries belong to the same parameter.

We agreed and added a thin horizonal line after each row to improve readability.

13. **P6, L136:** This last sentence belongs two the first sentence of this section. You should move it there. It will fit better.

Sentence moved (L89).

14. **P8, L190:** back → black

Fixed.

15. **P9**, **Fig. 2:** The transparency is almost not visible on a printed version. Fixed. We decreased the transparency (Fig 5).

16. **P9, L214:** Here you jump over six figures to show something in a figure, which is not yet introduced. Please avoid this.

We moved Section 2.3.2 and Section 2.3.3 below the TIR processing sections to better align the flow of reading.

17. P10, L247: Differences in Kelvin, not in °C

We appreciate the reviewer's attention to detail. We have carefully considered this and note that temperature differences are consistently reported in degrees Celsius throughout the manuscript for clarity and consistency.

18. **P14, Fig. 5:** ... and boxplots (b)  $\rightarrow$  ... and (b) boxplots; actually I think that only one of the figures is needed. They more or less show the same.

We agree and merged Fig. 5a with Fig. 4.

19. **P14, L300:** to or by 0.53°C?

Thank you for pointing this out we clarified that the RMSE for surface temperature decreased by 0.53 °C, down to 0.58 °C" (L259).

20. **P15, L320:** Fig. 6a and the same for Fig. 6b a few lines later Fixed.

21. **P15, L326:** ... higher-resolution (10 s) sea...  $\rightarrow$  ... temporally higher resolved (10 s) sea...

Fixed (L288).

22. P15, L338: better to use radiation instead of light.

We deleted this sentence.

23. **P18, Fig. 8:** Here, I have several things. (i) Uncommon Lat/Long values. I don't know how to interpret them. (ii) The colour bar is unintuitive. To me blue colours give the impression of lower snow depth, but it is the other way round. (iii) The scale ranges from 0 m to 0.6 m. Does it mean that there are parts with bare ice in the images?

We changed the axis unit to geographical coordinates. We adapted the color bar for more intuitive interpretation. We set the colorbar from 0 to 0.6 m for scaling, but the minimum snow depth proxy value is 0.01m (see histogram Fig. 9a) so no bare ice is present. However, we added a sentence referring to Fig. 9 in the image caption of Fig. 7 to clarify this.

24. P19, L 398: Missing brackets for the citation.

Fixed (L402).

25. **P19, 403:** skip "and a small fraction of clean snow." It will be said in Line 405.

Fixed.

26. P19, L405: QGIS?

Geographic Information System. We explained the acronym (L404).

27. **P19, L 408:** RGB and in the next line R/G/B

Fixed.

28. **P21, Fig. 10:** Not really introduced and not really needed. What is the aspect?

We have clarified the term "aspect" is tied to dune orientation (L383). We believe this figure is valuable as it shows the range and distribution of calculated irradiance values across the study area. While aspect itself may not be a primary focus, it provides context about solar exposure variability and the random orientation of snow dunes in this test site.

29. P21, L432: flight legs instead of flight lines?

Fixed to "flight legs" (L441).

30. **P23, L453:** can't → cannot

We deleted that sentence in the process of restructuring the manuscript.

31. P23, L469: Don't start sentence with an abbreviation.

Fixed (L495).

32. **P24**, **Fig. 12**: Would be helpful to add a name list to the single figures. Maybe as a fourth column left or right of the graph, were you just write (vertically) All, Sediment, and No-Sediment.

We agree and added the titles "entire field", "sediment" and "no Sediment" to each row in Fig. 8, 9 and 10.

33. **P25, L486:** ... area of 200 m. → is no area

Changed to "200 m x 200 m" throughout the entire manuscript.

34. **P26, L500:** e.g. → e.g.,

**Fixed.**

35. **P27, L557:** ... traps the light is a not so well expression. Try to avoid it.

We deleted that sentence in the process of restructuring the manuscript.

36. **P28**, **Fig. 14**: Add All, Sediment, No-Sediment to the figures.

**Done.**

37. **P28, L568:** ... the the ...

**Fixed.**

38. **P28, L575:** Missing space sign in front of citation. Appears three times more within this and the next paragraph.

**Fixed.**

39. **P30, L627:** viable → sustainable?

We deleted that sentence in the process of restructuring the manuscript.

40. **P32**, **Fig. A3**: x axis of graph b is incomplete.

**Fixed (Fig. A5).**

41. **P33, Fig. A4:** (a) and (b) are missing inside the graphs. However, graph (b) is not really needed. It is already well visible in (a).

Fixed. We deleted the second panel and adjusted the colors (Fig. A6).

**Reference:**

Wendisch, M., et al.: Atmospheric and Surface Processes, and Feedback Mechanisms Determining Arctic Amplification: A Review of First Results and Prospects of the (AC)3 Project, Bull. Amer. Meteorol., 104 (1), E208–E242, doi:10.1175/BAMS-D-21-0218.1, 2023.

**02 Review**

**General Comments**

This study presents an innovative UAV-based approach to mapping snow surface temperature and topography on Antarctic landfast sea ice. The authors provide detailed documentation of their UAV flights, ground-based measurements, and image processing steps, including a novel algorithm to correct thermal drift due to NUC (Non-Uniformity Correction) events in the thermal camera.

The study is technically sound and methodologically robust, offering high-resolution spatial insights into processes that are often oversimplified in energy balance models. The authors convincingly demonstrate that sediment and irradiance variability, rather than snow depth, dominate surface temperature patterns in their study area.

However, the manuscript in its current form is excessively long and includes some minor technical and formatting inconsistencies. Furthermore, the broader applicability of the results are not enough. With revision, this manuscript has strong potential for publication and contributes valuable observational data and methodology to polar remote sensing and surface energy balance research.

**Major comments**

1. My first major comments is the primary hypothesis, that snow depth drives spatial variability in snow surface temperature, is tested but ultimately unsupported. For example, the correlation between snow depth and surface temperature is weak (e.g., Spearman's rs = 0.16 for the entire field), which the authors acknowledge. While the discussion then shifts toward the influence of sediment and irradiance, the analysis remains largely qualitative.

In response to this valuable comment (among others) we rephrased our objectives towards a more general approach. Instead of saying we want to investigate snow depth as the key driver for surface temperature we changed it to a more general phrasing. Now we aim to understand the influence of different parameters (snow depth, sediment presence and micro-topography driven irradiance [L73]) on snow surface temperature variability. We restructured the manuscript accordingly by merging our Results and Discussion section (3) and now clearly discussing the three different parameters and their influence on snow surface temperature.

A more rigorous statistical treatment, such as a multivariate regression analysis that accounts for snow depth, sediment presence (e.g., red-band reflectance), irradiance, and roughness, would be necessary to better disentangle the relative contributions of each factor.

We added section 3.2.4 Integrating predictors via Linear Mixed-effects Modeling (L527). Here, we use the snow depth proxy, irradiance, and red band values as predictors of snow surface temperature. We combined the "sediment" and "no sediment" datasets and applied a linear mixed-effects model (LMM) to better quantify the relative influence of the different parameters on snow surface temperature. The LMM provides a formal, integrated evaluation of the drivers and enables a comparison of their practical impact

over the observed range of variability in surface temperature. The LMM confirms and quantifies the patterns observed in our correlation analyses. All three predictors - snow depth, irradiance, and red band intensity – were significant, with irradiance producing the largest potential change in surface temperature.

Moreover, the identification of sediment patches is based solely on manual inspection of RGB images and red band intensity. This approach is subjective and sensitive to camera auto-settings and changing illumination conditions. As a result, I have serious concerns about the accuracy and reproducibility of this classification. Could the authors develop an objective spectral threshold or a sediment index to quantify contamination levels? Could radiative transfer modeling or image normalization techniques help constrain uncertainty? How transferable is this method to other optical datasets or regions? As it stands, the interpretation of albedo-related heating effects remains qualitative and lacks the quantitative rigor needed to support strong conclusions. This weakens the scientific impact of the paper.

Firstly, we would like to clarify that the intent of the manual digitizing was not to rigorously identify sediment patches. Rather, it was to define two domains: one with significant amounts of sediment on the surface, but which also included clean snow, and secondly one with predominantly clean snow. This was necessary to untangle the competing influences of sediment and illumination in the optical imagery (red-band values). The delineation was straightforward to do manually – the sediment is clearly visible in the RGB imagery.

However, in a general sense we clearly see the reviewers point and agree that this approach for sediment detection in airborne red band orthomosaic is subjective and sensitive to camera auto-settings. We also agree that the reproducibility of this approach is limited. However, this study was somewhat exploratory, and the significant sediment presence in this apparently clean-snow area was unexpected. Consequently, our imagery was not designed to capture absolute surface brightness in the relevant spectral bands. The sediment presence is clearly a key driver of snow surface temperature which cannot be ignored – we believe that the approach in this paper is the best way that we could investigate this influence with RGB imagery.

We see our study as "opening the door" rather than providing a full analysis of all possible relations including EB modeling. So, in regard to the sediment detection process presented here, we want to clarify that future studies are in need of an automated sediment detection process, but this is not the focus of this study. Given the limitations of the available imagery (in this study), we do not consider the development of a more objective sediment index or the application of image normalization techniques feasible. However, for future work we are aiming to incorporate automated classification algorithms not only for sediment deposition but also the classification of different snow cover types based on the airborne products. For now, we acknowledge the subjectivity involved in manually delineating sediment patches. This study is intended as a proof of concept at a single site. Broader evaluation across different sites is planned for future work but lies outside the scope of the current paper. We hope this provides more context for our choices in this paper and outlines the bigger picture for future work.

2. While it is perfectly valid to report a null result (i.e., snow depth is not the main driver of surface temperature variability), the manuscript does not provide a compelling quantitative alternative. Sediment and irradiance effects are proposed as alternative drivers, but again, these are explored only descriptively, without any systematic modeling framework. There is no attempt to define a sediment contamination index or 2 perform a statistical regression that incorporates the full range of possible explanatory variables. As a result, the conclusions remain speculative and unsupported by robust evidence. This lack of quantitative follow-up significantly limits the paper's strength and broader relevance.

We agree that our initial focus on snow depth was misplaced, and have widened our hypothesis to include other potential drivers (L70). We respectfully disagree with the characterization that our analysis lacks a quantitative framework. While some relationships are indeed weak, this reflects the nature of the data rather than a lack of quantitative effort. We disagree that a weak correlation is not a result, because it shows that, for example snow depth, is clearly not the dominant driver of surface snow temperatures. Nevertheless, we agree on having an additional analysis to complement our pairwise correlation analyses. We included a LMM analysis incorporating the snow depth proxy, irradiance, and red band values as predictors of snow surface temperature (Section 3.2.4, L527). As previously noted, the development of a sediment contamination index lies outside the scope of this study. The revised manuscript includes additional analysis to support and clarify our conclusions.

3. My second major concern is the very limited spatial and temporal scope of the study, which raises questions about the broader significance and generalizability of the results. The entire analysis is based on a single 200 × 200 m region of landfast sea ice, under relatively flat and low-wind conditions. This kind of super-local case study might be sufficient for a technical demonstration, but how representative is it of more heterogeneous, dynamic and any other Antarctic sea ice environments? We are aware of the limited spatial and temporal scope of this study. This work includes a technical proof of concept for the retrieval of high-resolution temperatures, applied to a single test site under clear-sky and, low-wind conditions. We have also clearly and quantitatively shown (using a correlation of over 6000000 data points) in what is considered a relatively large coverage and high-resolution, of local in-situ surface temperatures over snow on sea ice, that microtopography plays a dominant role for local surface temperatures under clear sky conditions. This is a key takeaway message which should not be diminished by the (valid) needs for further research at different locations.

Can any of the findings, especially the role of sediment or microtopography in surface temperature variability, be extended to larger spatial scales or different surface types? What about temporal representativeness?

These are two different points which need to be clearly separated. We agree that the methods and analyses as well as results need to be applied and tested to different snow conditions/test fields. Not only to test for robustness but also understand the relevance of our findings in regard to different snow/ice types. So as for now we simply cannot say if our findings are applicable to a larger spatial scale, which clearly points out the need for future research. But our results are representative of snow cover on

relatively uniform sea ice - a major part of the sea ice in McMurdo Sound during that year.

The second point about the temporal representativeness is not easy to address on a broader scientific level. Antarctic field work is resource and time demanding due to the extreme remoteness and logistical nature. Generally, Antarctic fieldwork is conducted during the summer months from November to December. For this window our 4-week long study is representative in the annual context of that particular sea ice and snow cycle as we show the relatively stable weather conditions in our manuscript. But of course, it feeds into the general bias of summer measurements. There's almost no data about transition or winter months and we would like to help close that gap with future efforts.

The study captures only one moment in time under clear-sky conditions. Without any diurnal or seasonal coverage, I found the conclusions are difficult to generalize. Have the authors considered applying the same methods to a second site, or repeating flights under different light/wind conditions? If not, this should be clearly acknowledged as a limitation.

We show that it is possible to retrieve high-resolution temperatures (and topography and irradiance) at a single test site under clear-sky and low-wind conditions. We also clearly and quantitatively show (using a correlation of over 6000000 data points - what is considered a relatively large coverage and high-resolution of local in-situ surface temperatures over snow on sea ice) that microtopography plays a dominant role for local surface temperatures under clear sky conditions. It is also clear that this would not be the case under diffuse conditions, because of the lack of topography dependent irradiance. The use of the TIR camera is restricted to clear-sky periods, as clouds would force the surface temperature to reach thermal equilibrium with the cloud base temperature, overruling the impact of snow/ice spatial heterogeneity on the surface temperature map. While we recognize the value of expanding the analysis to additional sites or varying environmental conditions, such extensions are planned for future work and lie beyond the scope of this study.

4. Additionally, the snow depth proxy derived from the UAV DEM rests on the assumption of a flat ice surface and uses a single calibration offset from ground-based measurements. But where is the validation? There is no high-precision GNSS positioning of the MagnaProbe data, and the ±0.05 m uncertainty in snow depth is not propagated into any of the correlation analyses or surface temperature interpretations. How does this uncertainty affect the reliability of the snow temperature correlation results? Are there areas where DEM uncertainty dominates the signal? This aspect of the study is underdeveloped and undermines confidence in one of its core data layers. We agree that the snow depth proxy represents a potential source of uncertainty in our analysis. The approach relies on the measurement of a flat sea-ice surface underlying the snow and a single offset calibration based on 813 ground-based MagnaProbe measurement points, which do not have high-precision GNSS positioning. We included the propagated 0.045m error for the snow depth proxy in the LMM analyses (Section 3.2.4; Table 3) and can show that even adding this "noise" does not change the results. We have done the same thing for the surface temperature and added 2.41°C noise. We

think that this is now a sufficient approach (uncertainty propagation) to understand the impact of measurement uncertainty/noise on our results.

5. A further issue is the lack of an overall uncertainty framework. The authors had ±2.1°C for temperature correction (Figure 4), but there is little analysis of how this uncertainty varies spatially (e.g., near image edges, in shaded areas), or how it might interact with surface roughness, sediment detection, or irradiance calculation. What are the sources of uncertainty from lens effects, and how do they affect the final interpretation? Could they lead to systematic biases in warmer sediment zones or sloped surfaces?

We acknowledge this comment and want to point out three things.

- 1. The spatial variability of temperature retrievals is sampled where we assess the uncertainty with and without the NUC correction (in section 2.3.2). Each part of the TIR orthomosaic is covered by several overlapping images (frontal image overlap is 95% [L178] and side image overlap is 80% [L179]). We take measurements from the 250 sample sites where there are more than 10 images (on average 20). The images are captured by multiple passes in different directions, reducing the potential for systematic biases.
- 2. We apply an oval-shaped mask to all TIR images to eliminate temperature distortions in the images. This is included in Fig. 2 "Oval image mask applied to account for the vignette effect (corners of the image are cooler than the centre) in thermal images due to the lens". We also mention this in L264 as "oval-shaped vignette mask". The image edges were removed and not included in the dataset.
- 3. We now also include the propagated temperature error in the LMM analyses (Section 3.2.4; Table 3) and show that adding "noise" does not change the outcome.
- 6. There is also a surprising absence of any actual surface energy budget analysis. The paper frequently references "energy balance" but never attempts to break it down into its components (e.g., incoming solar, reflected solar, outgoing longwave, conduction). Without this, the proposed mechanisms, e.g., sediment warming or irradiance enhancement on slopes, remain less clearly. Why not use the DEM and albedo proxies to estimate slope-corrected irradiance per pixel, or combine reflectance and temperature into a basic radiative balance estimate?

Considering that we have calculated the slope-corrected irradiance per pixel and we discuss it in quite a bit of detail (e.g. Section 2.3.7, Section 3:Fig 10, Fig. 9, Fig. 11), we find this comment a bit puzzling. Our discussion involves the slope-corrected irradiance and its correlation with the measured surface temperature. We do not have albedo proxies, as our RGB measurements are not set up to retrieve albedo. Conducting a comprehensive energy balance breakdown is beyond the scope of this study, which is focused primarily on mapping spatial variability in snow surface temperature and exploring its potential drivers. Additionally, EB modelling would

require a lot of sensitivity analysis to input parameters and boundary conditions, to account for the large variation and uncertainties. We revised the manuscript to better frame the discussion around temperature variability and avoid overstating conclusions related to the full energy budget.

7. Finally, the overall discussion remains descriptive and lacks synthesis. What is the key physical message here? Are the findings consistent with prior studies in the Antarctic? What is new about the relationship between microtopography, sediment, and surface temperature, and what does it imply for satellite-based retrievals or sea ice models? Without clearer integration of results into a broader scientific context, the paper risks reading more like a high-resolution mapping exercise than a process-oriented scientific study.

Our study goes beyond descriptive mapping by using high-resolution UAV data to quantitatively link small-scale variations in snow surface temperature to potential controlling factors. Specifically, we integrate the snow depth proxy, red band intensity (as a proxy for sediment deposition), and modeled topography-dependent irradiance to identify the dominant drivers of surface temperature variability. Using a Linear Mixed-effects Model, we quantify the independent contributions of each factor, showing that sediment and local irradiance are the primary controls, while snow depth plays a secondary role. These results provide process-oriented insights into the relative importance of surface characteristics and illumination for near-surface energy balance, which can inform future high-resolution studies and improve parameterizations for sea ice models and satellite retrievals. We acknowledge the potential for tighter embedding of our findings into existing literature. However, such literature and datasets are very scarce, particularly regarding snow physical properties on landfast sea ice. Our intention is to first apply these methods to additional test sites to understand broader patterns of snow on landfast sea ice in McMurdo Sound during November 2022, and subsequently evaluate the representativeness of our current study.

**Specific comments:**

1. Section 2.3.3: The snow depth proxy construction is hard to follow. Are you aligning in situ snow depth data with the DEM by GPS coordinates, or using a 3 stereophotogrammetric offset? Please clarify the procedure with a schematic or simple diagram. This section needs better structure to explain how the reference elevation is selected and how uncertainty is estimated.

We use the minimum MagnaProbe snow depth and minimum DEM elevation to calculate a single offset between these two datasets and apply this offset to each pixel of the DEM. We then propagate the vertical error from the DEM (coming from the metashape routine) and the vertical error form the MagnaProbe. We added information on this in the section 2.3.6 Snow depth proxy (L310) and think it is now clear and not in need of a schematic. In our LMM analyses (Section 3.2.4) we include the propagated error for the snow depth proxy to acount for the uncertainty introduced by our method.

2. Figure 8: (1) The latitude/longitude labels are unconventional and difficult to interpret; (2) The color bar is unintuitive (blue implies deep snow, which is opposite of

expectations). Consider reversing the color scale or annotating it more clearly. (3) Does the 0 m value mean bare ice? Please explain this in the caption.

We changed the axis unit to geographical coordinates. We adapted the color bar for more intuitive interpretation (blue corresponds to shallower snow, green to deeper snow.). We set the colorbar from 0 to 0.6 m for scaling, but the minimum snow depth proxy value is 0.01m (see histogram Fig. 9a) - so no bare ice is present in this test site. However, we added a sentence in the image caption of Fig. 7 that refers to Fig. 9 for clarification.

3. Figures 10 and 13. These are not well-integrated into the discussion and don't provide substantial new insight. Consider removing them or clearly stating their purpose.

We improved the reference to Fig. 10 (now Fig. 11) in the revised manuscript and deleted Fig. 13.

4. Line 300:"...to/by 0.53 °C?". Please clarify whether this value is a change "by" or an absolute value "of".

We clarified that the RMSE for surface temperature decreased by 0.53 °C, down to 0.58 °C (L259).

5. Line 486: "...area of 200 m.". This is not an area. You probably mean "length of 200 m" or "area of 200 m  $\times$  200 m".

We corrected the phrase to "area of 200 m × 200 m" in the revised manuscript.

6. Line 332 and Line 568: remove one "the"

We removed the redundant "the" in both instances.

7. Figure A2 and A3 should be swapped.

We fixed the order of figures in the appendix.

8. Figure A4 (b) and (c) add little value, it's redundant with (a). Consider removing or merging.

We removed the second panel (Fig. A6).

**03 Review**

**General**

This article examines the spatial variations in snow surface temperature at the plot scale (hundreds of meters) on sea ice, aiming to identify the primary drivers. Snow temperature is highly sensitive to even minor changes in incoming fluxes (such as shortwave and longwave radiation) and serves as a means to explore the surface energy budget (SEB) and its unique characteristics on snow. To address this question effectively, the authors collected a unique and exceptional dataset in the Antarctic. The paper holds significant interest for the TC community but has some shortcomings, one of which is notable yet addressable.

The main issue lies in how the paper's question and objective are approached. The hypothesis proposed by the authors to justify and structure the study is unconvincing from my perspective, and they ultimately demonstrate that this hypothesis is indeed not validated. Instead, they identify one/two more adequate hypotheses, which, to my knowledge, are more obvious. While it is appreciable to build a paper around a clearly stated hypothesis, the fact that this hypothesis is not commonly supported in the literature and can be dismissed with magnitude order calculations undermines the paper's construction. What if the hypothesis had been, "Does local topography explain most of the surface temperature variations at the plot scale?" Research on the link between topography and SEB is prevalent in the literature, and the current title aligns more closely with this hypothesis. With the same dataset and results, the introduction and discussion would flow better, aligning more effectively with the results and conclusion.

Apart from this shortcoming, the paper is excellent and has many strengths, including significant work on TIR camera correction and an exceptional dataset. The paper is clear and well-written, with an easy-to-follow logic (except the order of the figures) and a generally sufficient level of detail (see the exception on the model below). However, the abstract is difficult to understand without having read the paper, as the logical progression is unclear (unlike in the paper itself). To attract more readers, I suggest rewriting the abstract from scratch, incorporating more results and fewer hypotheses.

The paper is quite long and my recommendation for the review is to shorten where possible, but avoid lengthening (except for the model description!).

We appreciate the reviewer's thoughtful comments and fully agree with the concerns raised regarding the framing of our hypothesis. We revised the manuscript to shift away from the narrow focus on snow depth as the primary driver of snow surface temperature variability.

Instead, we adopted a more comprehensive and better-supported list of our objectives at the end of the introduction starting in L70: "(1) To develop and validate an airborne method for accurately mapping snow surface topography and surface temperature on flat sea ice in the polar regions. (2) To use this airborne method to quantify the snow topography and surface temperature variability over landfast, flat Antarctic sea ice at the 100 m - scale. (3) To investigate the relative influence of potential drivers - including

snow depth, sediment presence, and microtopography-driven irradiance variations - on snow surface temperature. "

This re-framing is now also reflected in the merged Results and Discussion Section as well as in the rewritten abstract. We also acknowledged the comment regarding manuscript length and made targeted reductions to enhance clarity without compromising content. However, we expanded the (irradiance) model description section to ensure sufficient detail is provided for reproducibility and transparency. We also included a Mixed-effect Model analyses to add another layer of quantification to our pairwise correlation analyses. Finally, we changed the title slightly to better reflect the content of our manuscript, exchanging "roughness" with snow topography .

**Detailed comments:**

L 6 "Our airborne maps reveal a mean snow depth of  $0.16 \pm 0.06$  m". The mention of snow depth measurements is new in this sentence. The previous sentence is about surface temperature and the end of this sentence is about surface temperature. A reorganisation is necessary.

We restructured the entire abstract to improve logic and provide a better overview about the manuscript methodology and structure. We now mention the average snow depth from the MagnaProbe measurements (L8) rather than the DEM snow depth proxy results to not run into the problem of having to explain this complicated method in the abstract and provide the reader with an overview of the test site conditions.

L11 "seemingly flat snow field" can you give a number, this statement is relative to reader's expectation of what flat is.

By restructuring the abstract this sentence was deleted. We only refer to the uniformity of the sea ice which is backed up by the average thickness in L5: "(...) test site on relatively uniform landfast sea ice  $(2.4 \pm 0.04 \text{ m thick})$ ".

L13-14. It is not clear what the variability accounted in the uniform irradiance model. What is meant exactly by "the incoming solar radiation (irradiance) at the point scale". In principle incoming solar radiation is measured w/r to a horizontal surface. Do you mean the solar radiation received / perceived by the surface?

To clarify, our irradiance model computes the solar radiation actually received by each inclined sea-ice surface element, accounting for local slope, aspect, solar geometry, and terrain shading-rather than using irradiance values referenced to a horizontal surface. It combines ground-based measurements of global and diffuse radiation during the flight with the DEM-derived surface geometry using established radiative transfer approaches. We revised the manuscript and now thoroughly explain the irradiance model in "Section 2.3.7 Irradiance calculation" including the core equations for direct irradiance, diffuse radiation correction and slope- and shading-corrected surface irradiance.

L17 "While we initially hypothesized that snow depth was a key driver of snow surface temperature," this hypothesis should be stated in the first part of the abstract (L5) to position the problem addressed in the paper, and it should be justified, backed by literature because this hypothesis is not intuitive, at least to me. From the SEB equation, I'd expect surface temperature to depend on irradiance first, the snow depth does not appear in this equation unless the conductive term is written with the Fourier law. However even in this case, the snow is far too insulating to allow this term to become significant with respect to the others in summer, especially the incoming irradiance.

Our original hypothesis was motivated by the fact that, because of the strong insulative and scattering properties of snow, in case of shallow snow covers small changes in snow depth have a strong impact on the heat conduction and surface albedo. We have revised the abstract and introduction to frame our initial hypothesis more broadly - acknowledging that while we considered snow depth as one possible contributor to spatial variability in surface temperature, we also evaluated other controls, particularly topography, irradiance and sediment presence.

In detail in L70 we now clearly present the objectives of our study: "(1) To develop and validate an airborne method for accurately mapping snow surface topography and surface temperature on flat sea ice in the polar regions. (2) To use this airborne method to quantify the snow topography and surface temperature variability over landfast, flat Antarctic sea ice at the 100 m - scale. (3) To investigate the relative influence of potential drivers - including snow depth, sediment presence, and microtopography-driven irradiance variations - on snow surface temperature."

L34-35: "through the satellite period" "since record-keeping began". Should indicate the starting year to avoid ambiguity.

We added 1978 in the paragraph (L27): "In contrast, the minimum Antarctic sea ice extent exhibits more variability, with a general increase through the satellite period (since 1978) until 2016, and a decrease thereafter. Recent records show the lowest minimum extent since record-keeping began in 1978 (Wang et al., 2022)."

L47. While the authors are free to make the hypothesis they want as long as it is clearly stated – and I acknowledge it is very well done here w/r to the literature in general -- still I found this hypothesis strange. The physical reasoning behind this hypothesis should be developed a bit and examples from the literature could help.

We acknowledge the point of our hypothesis being very narrow and decided to instead of framing the manuscript around a hypothesis to just present our objectives and restructure the manuscript accordingly. Hence, this sentence was changed to "Despite the recognized influence of snow properties on sea ice thermodynamics, the relative importance of different factors - such as snow depth, surface microtopography, and sediment deposition - on snow surface temperature remains poorly understood." (L39).

L90. The objective would benefit to be rewritten without this simple hypothesis. A more neutral approach would be to list all the potential factors influencing the small scale variability of snow surface temperature with a short literature review for each, and

reframe the goal into investigating/quantifying which term is the key driver in the specific context of this study (summer, sea-ice with small snow depth).

We followed the reviewer's suggestion and rephrased the entire objective paragraph: "(1) To develop and validate an airborne method for accurately mapping snow surface topography and surface temperature on flat sea ice in the polar regions. (2) To use this airborne method to quantify the snow topography and surface temperature variability over landfast, flat Antarctic sea ice at the 100 m - scale. (3) To investigate the relative influence of potential drivers - including snow depth, sediment presence, and microtopography-driven irradiance variations - on snow surface temperature." Now, instead of centering the study around a single hypothesis, we explicitly list the potential factors influencing small-scale variability in snow surface temperature (including snow depth, sediment presence, and irradiance variations). This revision aligns the study objectives more closely with both the existing literature review and our dataset, and improves the logical flow between the introduction, methods, and results.

L116: "The other four sites" I'd remove this sentence, it diverts from the objective of the paper.

We deleted the sentence and rephrased in L108: "During the November 2022 field campaign, we collected data on five  $200 \text{ m} \times 200 \text{ m}$  sites with different snow and ice conditions, of which, in this study, we only discuss the site we assumed to be a "flat and relatively homogeneous" .

L136. Please check the correspondence between the height of installation vs the footprint areas (1.6m2 versus 0.35m2). Is the angle of installation at the sediment site different?

We confirm that the difference in footprint area is due to installation geometry: the clean-snow sensor was mounted at  $\sim$ 1.2 m height with a 45° tilt, resulting in an elongated footprint of  $\sim$ 1.6 m², while the sediment sensor was mounted at 1.02 m height pointing vertically downward (nadir), yielding a near-circular footprint of  $\sim$ 0.35 m². We clarified this explicitly in the manuscript (L126).

L214: Check that the figures are referred in order (it seems Fig 8 is referred before others).

We changed the order of images, and they are now referred to in the correct order.

**L215 L223: Check figure A1 reference**

We corrected the reference which now connects to Fig. A2 showing the smoothness of the sea-ice surface after removing the snow.

Figure 2. Can you show a scatterplot (+ r2 and RMSE) between the inferred snow depth proxy and the magnaprobe measurements as a validation of the approach? We have generated the suggested plot (Fig. A3), which shows that a point-to-point correction of the DEM using MagnaProbe measurements is not feasible. The spatial variability of snow depth within the GPS uncertainty footprint (r = 2.5 m) is too high for such a correction to be meaningful. We included the offset-corrected and offset-uncorrected DEM elevation as well as the MagnaProbe measurement for each of the 6 randomly chosen sites (Fig. A3) on the MagnaProbe measurement transect.

L272: It is not obvious how 0.5 °C was found based on Fig 4a. I guess it is empirical but how sensitive to conditions is it?

We clarified how the threshold was identified in L226: "NUC events in this study were identified using a fixed threshold of 0.5 °C, chosen empirically based on the stability of the correction curve (Fig. 3a). We tested different thresholds (1 °C, 0.5 °C, and a dynamic 99th percentile approach) and found that 0.5 °C visually identified the NUC events most clearly and consistently at this site. "For sites with more heterogeneous temperature conditions, we are going to define the threshold dynamically as the 99th percentile of the temperature difference distribution, which makes the detection adaptive to local variability.

**L318: by curiosity, how large is this correction?**

We interpret the reviewer's question as referring to the magnitude of the correction between the Apogee brightness temperature and the derived surface temperature in Eq. 4. For the Apogee pointed at the sedimented snow surface, the mean correction is 0.55 °C.

L321: I don't understand what this RMSE is, between what and what it is calculated (+ typo: with an RMSE => with a RMSE)

L283: We corrected the typo and elaborated as following: "with a RMSE of 0.58 °C quantifying the residual error in the temperature dataset and serving as an uncertainty estimate for the surface temperature maps".

L329: "The RMSE of the residuals of this linear fit". RMSE → RMS or remove residuals L292: We changed the wording to: "The total uncertainty of the derived surface temperature is calculated as the square root of the sum of the following squared contributions: the RMSE of the linear fit (...)".

L330. "the square of the thermal" and "the square of the RMSE associated " check if square is correct in both cases. I'd recommend to completely rephrase or write as an equation.

L292: We rephrased the paragraph to explicitly state the uncertainty contributions from the linear fit, thermal camera, and the RMSE associated with the NUC correction (see response to comment above).

Section 2.3.8. It is not clear how the impurities are detected. Using a threshold or just visually? If just visually, this section could be removed, and a line or two in the results section is sufficient.

We removed Section 2.3.8 and incorporated a concise description of this step in the Results and Discussion section. We shifted the description to L372, when introducing the red band orthomosaic and elaborated on the visual delineation of sediment patches in L404 when discussing the role of sediment.

Section 2.3.9. Given the critical role of the model, the level of detail should match that given for the drone. It is necessary to provide the main aspect of the model (e.g.

workflow) and present the equations or refer to the equations in the cited papers for each main calculation step.

Main questions are: the diffuse component, the resolution of the calculation (in relation with the positioning uncertainties), cast shadows, multiple scattering (esp for the shadows).

We revised Section 2.3.9 (now 2.3.7). We included a clearer description of the model workflow including the core equations and measurements used for the irradiance model (Eq. 6-8, L350 - L363). In this we include treatment of the diffuse component (directly using the measurement recorded by the SPN1 sensor, clarified at L347). The resolution of the calculation is 9 cm, the pixel size of the DEM, which is comparable to the position uncertainty. Only a very small part of the field is in cast shadow, but this is calculated explicitly (shaded cells are masked and receive zero direct radiation (L353–354, Eq. 8)). We do not consider multiple scattering and simply apply the diffuse component (with the required spectral scaling) to shadowed areas (L357 - L364).

L360. How does  $2.4 \pm 0.04$  m translate into 1% variation and where 0.04m is coming from ? The histogram seems to indicate larger deviation. How relative variation is defined and calculated ?

The value of ±0.04 m refers to the standard deviation of all 2449 sea ice thickness measurements and was not used for calculating the relative variation. The 1% variation was derived from the mean absolute deviation (MAD) of the thickness values, expressed as a percentage of the mean thickness (2.4 m). We rephrased this in L335: "The relative variability in sea ice thickness was calculated as the mean absolute deviation of all measurements expressed as a percentage of the mean thickness, yielding 1 % (Fig. 6). Thickness ranged from 2.28 m to 2.52 m with a measurement uncertainty of 0.1 m (Table 1)."

L395. Fig 12 is referred, check the order.

We restructured the manuscript to ensure all figures are referenced in order.

L413. Isn't it due to the correlation between snow depth and impurities?

We thank the reviewer for the suggestion. However, we do not think the stronger correlation is due to a direct relationship between sediment and snow depth. The "sediment" dataset includes outlines drawn around sediment patches, but these masks also include surrounding clean ice pixels, introducing greater variability in both snow depth and temperature. In contrast, the "no sediment" dataset is based on rectangular boxes placed in uniform snow areas, resulting in more homogeneous conditions and a narrower temperature range. Therefore, we attribute the stronger correlation in the "sediment" dataset to this broader variability rather than a direct link between sediment cover and snow depth. The interrelationships between sediment cover and snow thickness are explored in the new section 3.2.4

L469. Why is this relevant in this section? It is well known that the irradiance depends on the local incidence angle, and not on the slope.

We agree that irradiance strictly depends on the local incidence angle, not slope alone. However, in this section, we refer to slope as a proxy for estimating how the local

surface orientation influences the irradiance received - i.e.," tilted irradiance". We acknowledge that the term "irradiance" may have been used imprecisely here. We clarified the language in L495: "Slope serves as a proxy for surface orientation, capturing variations in the local incidence angle and, consequently, in effective irradiance."

L476: The aspect distribution is uniform, not gaussian. The slope distribution is not Gaussian.

L501: Corrected the wording to "uniform distribution".

Figure 12. For convenience, adding titles on the rows directly in the graph would help quickly read the figure, without having to read the caption.

The x-axis scale is very large, for just a few outliers. I suggest to reduce the range to - 17°C - -10°C or so. It would make the graphs H and I more convincing for instance. We added the dataset titles ("entire field", "sediment", "no sediment") above each row in Fig. 9. and limited the x-axis temperature range down to -18 to -8°C.

L491: I'm not sure but I think that the algorithm not only correct for NUC jumps but also for other trends in the camera which are usually very large.

Note that other cameras do a "better" job in NUC smoothing which makes jumps more difficult to detect... while still be necessary to applied the necessary corrections. See for instance: Arioli, S., Picard, G., Arnaud, L., Gascoin, S., Alonso-González, E., Poizat, M., and Irvine, M.: Time series of alpine snow surface radiative temperature maps from high precision thermal infrared imaging, Earth Syst. Sci. Data, 16, 3913–3934, doi: 10.5194/essd-16-3913-2024, 2024

Ideally, one would access the raw data... but it seems that camera manufacturer prefer to overprotect their (insufficient) algorithms.

Thank you for pointing this out and for the great reference. We agree that our correction algorithm is relatively simple and may not capture all forms of sensor drift or be directly transferable to all camera models, especially those with more complex or opaque NUC behavior. Still, we see it as a useful baseline that demonstrates the value of transparent correction methods and offers a starting point for adaptation in other environments and systems. Importantly, this correction was necessary to make the thermal data usable at all - without it, the images could not have been processed into an orthomosaic without much larger uncertainty (section 2.3.2, L253).

L500. I'd advocate for m ore accurate sensors than Apogee sensors when absolute value is important (e.g. close to 0°C, see Arioli et al. 2024)

We agree with this statement. A colleague of ours has developed a really nice method of large metal plates (painted with a black paint of known emissivity) on top of an insulative foam and a thermocouple that logs the temperature of the plate. We plan to use that in the future.

L539. "While the red band values do not directly affect the surface energy balance, we use them as a proxy for impurities." I don't understand this statement. Maybe the verb "affect" is incorrect.

We agree the original phrasing was unclear. Red band values alone cannot be used to infer surface impurities for surface energy balance calculations (and albedo), in our case, they serve as a proxy for surface impurities (and potential energy absorption by the impurities). But we acknowledge that the red band values are influenced by illumination as well by impurities, as discussed throughout the manuscript. This sentence was deleted in the process of restructuring the manuscript.

L550. I would suggest to coarsen the resolution a bit to account for the positioning uncertainty and to see how this correlation increases. Mathematically, the correlation always increases with smoothing, but here the idea is to see how quick it increases. We have added Figure A3, which shows the variability in surface elevation around 6 randomly-selected MagnaProbe locations. The takeaway is that the variability in the DEM-derived elevation/snow depth proxy within the radius around the MagnaProbe is almost as much as in the whole field. The reviewers suggestion would lead to the same conclusion – coarsening the resolution would sample from the area around the MagnaProbe and would simply average out this variability into a mid-range value and not provide a better correlation. Unfortunately, without precise positioning of the MagnaProbe measurements we cannot achieve a better correlation.

L567: the the => the Corrected.

L570: while this result is sound, the statistical demonstration would require first to demonstrate that topography and impurities are independent in your case. For instance if the impurities areas had more north looking slopes, the relationship is biased. It is frequent (in the mountains) that the sun facing slopes are more likely to have dust emerging at the surface than the colder faces.

We appreciate the comment. However, in our sea ice setting, sediment is likely wind-deposited rather than emerging from the snowpack due to melt, as is common in mountainous terrain. That said, we do observe some correlation between sediment presence and topography, with sediment-covered areas tending to be higher and in areas of thicker snowcover. These effects are discussed briefly at L415, but the interrelationships between variables are also considered in the MLM in section 3.2.4.

L580 I suggest to also mention multiple scattering which is likely important in the cast shadows areas and cavity effects in the LW which is probably negligible with slopes <10°. Ref: A. Robledano, G. Picard, L. Arnaud, F. Larue, I. Ollivier, Modelling surface temperature and radiation budget of snow-covered complex terrains, The Cryosphere, 16, 559–579, doi:10.5194/tc-16-559-2022, 2022

We thank the reviewer for highlighting the potential role of multiple scattering. We agree that multiple scattering can enhance irradiance in complex terrain. However, our study site on landfast sea ice exhibits shallow slopes and a relatively uniform snow surface, so we consider this effect to be minor. We have clarified this assumption in the revised manuscript (L364).

L608: This is not necessarily a drawback. If only the irradiance is changing (not Tair, not wind), observing two different Ts give a lot of information on the balance between SW and the other terms of the SEB.

AC: We thank the reviewer for this insightful comment. The two flights were necessary to acquire the RGB data for constructing the DEM, and we combined the thermal infrared data from both flights to generate a single, averaged orthomosaic for this study. For future work, we are considering analyzing the flights separately or using just one flight to build the orthomosaic, which could allow investigation of surface temperature changes related to varying irradiance.

L630: "offering valuable tools for many users". Same comment as before. To my experience, using more expensive cameras with better NUC correction make the proposed solution not applicable and the problem more severe... A recommendation could be to buy or develop open-source cameras or at least cameras that have been evaluated by others and for which correction algorithms exist.

This is a very good point, and we appreciate the reviewer pointing out that better cameras may make this problem even worse. So, we are pleased to provide a solution to this relatively "inexpensive" off-the-shelf DJI setup, which has proven to reliably work in polar regions. The heated DJI batteries used with our camera enable sufficient flight durations to cover our test fields (200 m × 200 m), which remains challenging with non-heated alternatives. Many custom or alternative cameras face limitations such as insufficient battery life under the low temperatures typical of Antarctic conditions, but it's a good idea for the community to keep looking for open-source alternatives. With our ongoing research we are going to test this method on different sites and will include recommendations based on these findings.